

# Design and Characterization of a new OFR: The Particle Formation Accelerator (PFA)

Ningjin Xu[1,2], Don R. Collins[*1,2]

[1]Department of Chemical and Environmental Engineering, University of California Riverside,  Riverside, CA 92521
[2]College of Engineering- Center of Environmental Research and Technology (CE-CERT), University of California Riverside, Riverside, CA 92507

*Correspondence to*: Don R. Collins (donc@ucr.edu)

**Abstract.** Oxidation flow reactors (OFRs) are frequently used to study the formation and evolution of secondary aerosol (SA) in the atmosphere and have become valuable tools for improving the accuracy of model simulations and for depicting and
accelerating realistic atmospheric chemistry. Driven by rapid development of OFR techniques and the increasing appreciation of their wide application, we designed a new all-Teflon reactor, the Particle Formation Accelerator (PFA), and characterized it in the laboratory and with ambient air. A series of simulations and experiments were performed to characterize: (1) flow profiles in the reactor using computational fluid dynamics (CFD) simulations, (2) the UV intensity distribution in the reactor and the influence of it and varying $O_3$ concentration and relative humidity (RH) on the resulting equivalent OH exposure
($OH_{exp}$), (3) transmission efficiencies for gases and particles, (4) residence time distributions (RTD) for gases and particles using both computational simulations and experimental verification, (5) the production yield of secondary organic aerosol (SOA) from oxidation of α-pinene and *m*-xylene, (6) the effect of seed particles on resulting SA concentration, and (7) SA production from ambient air in Riverside, CA, U.S. The reactor response and characteristics are compared with those of a smog chamber (Caltech) and of other oxidation flow reactors (the Toronto Photo-Oxidation Tube (TPOT), the Caltech
Photooxidation Flow Tube (CPOT), and quartz and aluminum versions of Potential Aerosol Mass reactors (PAMs)).

Our studies show that: (1) $OH_{exp}$ can be varied over a range comparable to that of other OFRs, (2) particle transmission efficiency is over 75 % in the size range from 50 to 200 nm, after minimizing static charge on the Teflon surfaces, (3) the penetration efficiencies of $CO_2$ and $SO_2$ are 0.90 ± 0.02 and 0.76 ± 0.04, respectively, the latter of which is comparable to estimates for LVOCs, (4) a near laminar flow profile is expected based on CFD simulations and suggested by the RTD
experiment results, (5) *m*-xylene SOA and α-pinene SOA yields were 0.22 and 0.37, respectively, at about $3 \times 10^{11}$ molec. $cm^{-3}$ s OH exposure, (6) the mass ratio of seed particles to precursor gas has a significant effect on the amount of SOA formed, and (7) during measurements of SA production when sampling ambient air in Riverside, the mass concentration of SA formed in the reactor was an average of 1.8 times that of the ambient aerosol at the same time.



## 1 Introduction

Atmospheric aerosols play major roles in air pollution, global climate change, and visibility reduction (Seinfeld and Pandis, 2006). The complex mixtures of inorganic and organic species present in atmospheric aerosols originate from both direct, or primary, emissions and production of secondary aerosol (SA) from atmospheric reactions.  Organic aerosol (OA) makes up a substantial fraction of atmospheric aerosols, and is comprised of primary OA (POA) that is directly emitted in the particle phase and secondary OA (SOA) that is formed in the atmosphere through reactions of gas phase precursors.  SOA

forms when reaction of volatile organic compounds (VOCs) with gas phase oxidants produces less-volatile functionalized compounds (Pankow, 1994; Jimenez et al., 2009; George and Abbatt, 2010) and when water-soluble organics dissolve in the aqueous phase (aerosol water or cloud droplets) and are subsequently oxidized (Lim et al., 2010; Ervens et al., 2011). However, the mechanisms of SOA formation are still poorly understood and are continuously extended and refined. Part of the complexity of SOA formation arises from the numerous oxidation reactions involving the large number of VOCs in the

atmosphere (Aljawhary et al., 2016). Additionally, after formation from precursor gases the SOA can evolve through multiphase and multi-generational processes, forming more complex distributions of compounds comprised of thousands of molecules (Xu et al., 2015; Chen et al., 2018; Shrivastava et al., 2019).

For decades, comprehensive laboratory studies on the sources, formation, and aging of SOA have been conducted in batch-mode atmospheric reactors, which are also known as environmental chambers or smog chambers (Pandis et al., 1991;

Carter et al., 2005; Song et al., 2005; Weitkamp et al., 2007). Though such chambers can create environments that closely simulate the complexity of the atmosphere, results from their use are affected by the loss of particles and semi-volatile compounds to the walls (Zhang et al., 2014; Nah et al., 2016, 2017), by limitations on experiment duration and extent of reaction, and by potential leakage of room or outside air into the Teflon bags (Pierce et al., 2008; Matsunaga and Ziemann, 2010; Krechmer et al., 2015). Moreover, the size of common smog chambers, which typically range from about 5 to 250 $m^3$

(Lonneman et al., 1981; Mentel et al., 1996; Wang et al., 2014; Tkacik et al., 2017), imposes a substantial restriction on their use for studying aerosol formation in ambient air (Bruns et al., 2015).  Oxidation flow reactors (OFRs) have been developed as a complement to traditional smog chambers and offer advantages such as providing oxidant exposure that can greatly exceed that possible in smog chambers and is variable over a wide range (Kroll et al., 2009), portability for use in the field (Wong et al., 2011), and the ability to investigate time-varying sources of SA (Kang et al., 2007). Inside an OFR, extremely high

concentrations of hydroxyl radical and/or other oxidants are maintained (up to $10^{10}$ molec. $cm^{-3}$ for OH), such that sampled air experiences the equivalent of several hours to days or even weeks of oxidative chemistry over the residence time of just a few minutes. Shorter residence times minimize interaction of the gases and particles with walls (Keller and Burtscher, 2012) and permit measurements of dynamic environments and sources. The portability and flexibility of OFRs also make them versatile, with the same experimental system applicable for a variety of laboratory and field measurements. Their fast response also

makes them better suited than smog chambers for experiments probing the influence of a matrix of parameters on SOA formation (Slowik et al., 2012; Palm et al., 2018).


Previous investigations of potential SA formation using different types of OFRs have resulted in optimized designs and strategies for studying specific sources or processes, such as measuring time-resolved SOA formation from gasoline vehicles during a transient driving cycle (Karjalainen et al., 2016) and from rapidly changing vehicular emission sources

(Simonen et al., 2017). Several groups have employed OFRs to study SA formation from ambient air, with examples including investigation of the variability of precursor gases and the resulting SOA in a ponderosa pine forest (Palm et al., 2016), high time resolution quantification of SOA formation from ambient air in central Amazonia (Palm et al., 2018), and observation of SOA formation and aging from urban air (Ortega et al., 2016). For laboratory-based studies, the concentrations and SOA yield (Y) for certain precursors can serve as a reference to estimate total SOA potential (Iinuma et al., 2004; Loza et al., 2014).

Yields determined for common precursors can also provide a quantitative measure of performance of a reactor relative to others of varying design and purpose (Kang et al., 2011; Lambe et al., 2015). Numerous studies have been conducted investigating differences in the SOA yield between OFRs and large environmental smog chambers (Matsunaga and Ziemann, 2010; Bruns et al., 2015; Lambe et al., 2011, 2015). For example, Lambe et al. (2011) showed that the SOA yield they observed in a PAM reactor is similar to that reported for the Caltech smog chamber. Other studies focused on SOA produced from

different precursor gas sources. For example, Li et al. (2019) used a new OFR (the Environment and Climate Change Canada OFR; ECCC- OFR) to evaluate the SOA yields of single compounds (alkanes and α-pinene) and of complex precursor mixtures, such as emissions from oil sands. Ahlberg et al. (2017) found that using single precursor experiment yields could lead to underestimated SOA mass loadings if aerosol dynamics is not properly accounted for. Cubison et al. (2011) characterized the evolution of laboratory biomass burning emissions using a PAM reactor and Kang et al. (2011) estimated the

SOA-forming potential of model organic compounds.

Reactor design is a critical step in the development of an OFR system and determines overall applicability and performance. The geometry and dimensions of the reactor have substantial impacts on velocity profiles, residence time distributions, wall effects, and extent of reaction. The reactor design mainly includes the selection of materials, the inlet configuration, the diameter-to-length ratio, the body length, the strategies for mixing the reactants, and the mode(s) of

generating the hydroxyl radical or other oxidant(s). For example, some inlet designs can lead to dead zones near the reactor walls, increasing the difficulty of achieving laminar flow in the entrance of the reactor and broadening the RTD (Mitroo et al., 2018). Selection of wall materials and any surface treatments is guided by an application-dependent balance of the importance of loss of gas-phase compounds or delays in their transfer, loss of charged particles to non-conductive materials, and UV transmittance for designs for which the lamps are outside of the reactor. Common materials used in OFRs include chromated

aluminum (e.g., PAM), silicon-coated stainless steel (e.g., TPOT), and quartz (e.g., CPOT). Recent studies of organic gas transmission through common tubing types described by Deming et al. (2019) suggest PerfluoroAlkoxy (PFA) and Fluorinated ethylene propylene (FEP) Teflon may be alternative choices for applications for which minimizing wall losses of gases is a priority. The position and power output of the UV lamp(s) are determined by the reactor materials and their transparency and by temperature control requirements during operation (Kang et al., 2007; Ezell et al., 2010). With most OFRs, the lamps are



either mounted on the inner surface for metal-wall reactors or outside for quartz-based reactors. The emitted wavelengths and intensity uniformity of the UV lamp(s) are also important considerations in reactor design (Li et al., 2015).

Here we present the design of a new all-Teflon OFR and report the results of computational, laboratory, and field studies through which it was characterized. UV intensity and total OH exposure ($OH_{exp}$) were quantified inside the flow tube. The flow profile in the OFR was modeled and the resulting residence time distributions of gases and particles were both
modeled and experimentally verified. Two precursor species were used to investigate SOA yield and the dependence of that yield on variations in parameters such as precursor concentrations, OH exposure, and the presence and concentration of seed particles. SOA mass yields are compared with those reported in the literature for the same VOCs. Field testing was conducted by measuring SA formation in ambient air sampled in Riverside, CA, U.S. Collectively, these tests confirm the utility of the PFA reactor for both laboratory and field studies.

## 2 Experimental

### 2.1 Reactor design

A cutaway view of the Particle Formation Accelerator (PFA) OFR is shown in Fig. 1. It consists of a PFA-Teflon reactor tube sealed between inlet and outlet end caps that were machined from blocks of polytetrafluoroethylene (PTFE). The reactor is oriented vertically, with the inlet at the bottom and outlet at the top. The PFA tube has a volume of 7.5 L and
dimensions of 151 cm L × 7.8 cm ID. Both end caps have an OD of 10.2 cm and are sealed with the reactor tube by FEP-encapsulated O-rings. The small diameter-to-length ratio of the reactor section of 0.052 was selected to result in a narrow residence time distribution in the flow tube and a more uniform OH exposure (Lambe et al., 2011). However, the small diameter also results in a reactor surface area to volume ratio of 0.53 $cm^{-1}$ that is higher than that of the TPOT (0.33 $cm^{-1}$) and PAM (0.23 $cm^{-1}$).

Some design elements of the PFA reactor are similar to those of other recently developed OFRs (Kang et al., 2007; Karjalainen et al., 2016; Huang et al., 2017), though there are some important differences as well. The inlet end cap has a 5.1 cm L × 1.3 cm ID bore used as the main sample air injection port, two side injection ports for introducing seed particles and $O_3$, and a cone-shape diffuser. That cone, which serves as the transition between the inlet injection port and the reactor tube, has an angle of 35 degrees. The sample flow gradually expands and is expected to be fully developed shortly after entering the
reactor tube. A single length of PFA tube (Ametek FPP P/N 33HPSC40x3.00) is used as the main body in order to simplify construction. Only the central ~50 % of the flow through the reactor is extracted and analyzed. That sample flow converges through an exit cone in the outlet end cap that tapers at an angle of 24 degrees from an ID of 2.1 cm to the 0.33 cm ID of the outlet bore through the top of the end cap. The outer ~50 % of the flow that is most influenced by interactions with the reactor walls flows outside of the exit cone and into an annulus, where it is pulled through 12 uniformly spaced ~0.15 cm ID pinholes
and purged by a vacuum pump.

The outlet end cap has NPT thread ports to accommodate one or two 0.5 cm OD lamps. For the results discussed here, one 5.1 cm L × 0.5 cm OD ozone-free low-pressure mercury lamp (BHK. Inc; PN 80-1057-01) emitting 254 nm UV radiation was used. The handle of the UV lamp is secured and sealed with a Swagelok male connector fitting. Use of a relatively small and low power lamp at one end of the reactor is perhaps the most significant design difference between the PFA reactor

and other OFRs. One objective of the approach was to promote thermal stratification caused by the hot lamp at the top of the reactor in order to minimize convective mixing. An obvious complication is that UV intensity, and therefore OH production, is expected to decay with distance through the long reactor tube. To mitigate that decay, materials were selected that are highly UV reflective, such that emitted photons penetrate far down the reactor tube as they are repeatedly reflected by the walls. The PFA-Teflon tube is non-absorbing at 254 nm but is not opaque and would allow UV to leak out. Thus, the tube is wrapped

with an inner layer of highly reflective 0.32 cm thick expanded PTFE gasket (ePTFE; Inertech) and an outer layer of aluminized Mylar (Vivosun). Though the combination of materials results in sufficiently high reflectance for the 254 nm emission peak of a mercury lamp. Silva et al. (2010) showed that the reflectance of ePTFE at 175 nm is significantly lower, with the difference thought to be due to absorption by $O_2$ trapped in pores. Reflectance at the 185 nm emission peak of a mercury lamp is expected to be slightly higher than that at 175 nm, but it is likely that a significant intensity gradient would still exist and so a 254 nm-

only lamp is used and ozone generated externally and introduced with the sample flow. The high reflectance of the ePTFE at 254 nm directs UV back into the reactor tube and results in increased intensity and uniformity.

The reactor assembly is protected by a shell made from 13 × 13 cm square aluminum tube. Two U-bolts mounted through the surface of the aluminum shell hold the reactor securely, preventing accumulation of static charge that could otherwise result from shifting between the reactor body and the ePTFE and Mylar layers. The shell also provides a barrier to

reduce the accumulation of static charge from inadvertent touching or other contact. A total of four fans are mounted on opposite faces near the top and bottom of the shell. The fans near the bottom bring air into the space between the reactor and the shell and those near the top exhaust it, which works to remove heat generated by the UV lamp and to weaken the temperature gradient through the whole system. The average working temperature for the tests reported below was approximately 23.6 ℃, which is close to the room temperature of 22.7 ℃. Continuous operation for 6 hours resulted in a

temperature rise of less than 2 ℃.

## 2.2 Experimental setup

As noted above, the PFA is an OFR254-type oxidation flow reactor, in which $O_3$ must be generated externally and introduced with the sample flow (Li et al., 2015). The OH radicals are produced as the 254 nm UV radiation generates excited oxygen atoms, $O(^1D)$, and $O_2$ from the photolysis of $O_3$, and then the oxygen atoms react with $H_2O$ in ambient air or humidified

laboratory air. For the laboratory experiments described here, $O_3$ and humidified zero air were mixed with the tracer or precursor gas(es) prior to being introduced into the reactor inlet. The schematic of the PFA and associated experimental equipment is presented in Fig. 2. Ozone was produced by flowing zero air through an $O_3$ generator (Jelight Company Inc. Model 610,). The flow rate was controlled to 0.4 L min$^{-1}$ and the $O_3$ mixing ratio was monitored by an $O_3$ analyzer (Teledyne



Model T400U). The resulting $O_3$ concentration can be easily and precisely adjusted by changing the position of a sleeve that
covers a portion of the UV lamp or by adjusting the flow rate of air through the generator. Here we present two different setups
for laboratory and field experiments, which are shown in Fig. 2 (a) and (b), respectively. When used, seed particles were
generated using an atomizer and differential mobility analyzer, DMA, as is described in the RTD experiment section. The flow
at the outlet of the reactor was split using a Swagelok tee. From one leg of the tee a 150 cm L × 0.635 cm OD PFA tube was
connected to gas measurement instruments including the $O_3$ analyzer, an $SO_2$ analyzer (Teledyne Model T100UP), and a gas
chromatograph with flame ionization detector (GC-FID, SRI Inc. Model 8610C). A 0.95 cm OD stainless tube was connected
to the other leg of the tee and carried the aerosol exiting the reactor to a fabricated scanning mobility particle sizer (SMPS),
which measured the particle size distribution roughly once every 4 min. For the ambient air experiments, outdoor air was
brought inside the lab and to the PFA reactor with a 200 cm L × 0.95 cm OD anti-static PFA tube (Fluorotherm H2 PFA). A
cm length of copper tube was used as a bypass in parallel with the OFR, with sampling alternated between the two through
the use of an automated 3-way valve. Instrument operation and experimental sequencing were controlled using National
Instruments LabVIEW software.

The total flow rate for the laboratory tests was 3.5 L min$^{-1}$, corresponding to an average residence time of 130 s, while
those of the PAM, TPOT, and CPOT are about 100, 110, and 1500 s, respectively. A purge flow rate of 1 L min$^{-1}$ was extracted
from the annulus outside of the sample exit cone as described above. The ambient experiments were conducted using a slightly
lower flow rate of 3 L min$^{-1}$, resulting in a residence time of 150 s, with a 1.5 L min$^{-1}$ purge flow.

## 2.3 Flow dynamics characterization

To characterize the flow field and velocity distribution profile inside the PFA reactor, computational fluid dynamics
(CFD) simulations were performed using a 3D geometry model in COMSOL Multiphysics 5.4 software. Several research
groups have used COMSOL to optimize and evaluate their reactor designs and to explore suitability for applications in
atmospheric and aerosol chemistry studies (Renbaum-Wolff et al., 2013; Zhang et al., 2015; Huang et al., 2017). Here, we
coupled the Laminar Flow and the Transport of Diluted Species built-in modules and used a finite element method to simulate
specific experimental conditions. The simulation geometry includes a diffuser inlet, reaction section, and cone outlet, all with
exact dimensions. A finer mesh comprised of approximately $3.59 \times 10^6$ tetrahedral elements was applied to the entire three-
dimensional model to capture flow dynamics near the entrance to the exit cone. An impermeable and no-slip boundary
condition was applied to all surfaces. The inlet flow to the reactor was prescribed to be 3.5 L min$^{-1}$ and the outlet pressure was
assumed to be atmospheric. The simulations were run until a steady state was achieved and the errors converged to $< 10^{-5}$.

## 2.4 RTD experiments

The residence time distributions of particles and gases were experimentally characterized and compared with results
obtained from an ideal laminar flow model simulation. The experimental configuration is illustrated in Fig. 2 (a). Monodisperse
ammonium sulfate (AS) particles were generated by atomizing a 0.04 M aqueous AS solution with an atomizer (TSI Inc.





Model 3076). The atomized particles were dried by directing them through a silica gel/ molecular sieve diffusion column. The size of the particles was selected using a differential mobility analyzer (DMA). The aerosol was brought to a steady state charge distribution before and after size classification by the DMA using soft x-ray neutralizers. The residence time distributions (RTDs) for particles were characterized by introducing 30 s pulses of 200 nm AS particles into the PFA reactor
while measuring the particle counts in the outlet flow with a condensation particle counter (CPC, TSI Inc. Model 3762)

        RTDs for gases were characterized by injecting 10 s pulses of $SO_2$ and $CO_2$. Pulses of a compressed gas mixture containing 27.5 ppm $SO_2$ in nitrogen (Airgas) were injected into a continuous zero air flow, with the pulse width controlled by opening and closing a mass-flow controller (Alicat Scientific, PN MC-100SCCM-D/5M). The $SO_2$ concentration was monitored from the sampling outlet of the PFA reactor with an $SO_2$ analyzer. Prior to the measurements, the reactor was purged
with zero air for as long as required to reach a measured $SO_2$ mixing ratio that was stable at less than 0.5 ppb. To test the response function of a gas that would not react on or be taken up by the walls, 10 s pulses of $CO_2$ were injected from a custom-made $CO_2$ tank, with the pulses controlled by manually opening and closing a valve. The $CO_2$ concentration was measured at the outlet of the PFA reactor by a $CO_2/H_2O$ gas analyzer (Li-COR Biosciences, Model Li-840A). A $CO_2$ background of 400 ppm was subtracted from the results because it was not removed by the zero air generator. The residence time distributions of
both gases and particles were determined with the UV lamp turned on and turned off. The whole process described above was repeated three times.

**2.5 Gas and particle loss quantification**

        Particle losses in the reactor were characterized using AS particles within the diameter range from 50 to 200 nm. The monodisperse AS particles were size-selected by a DMA and then passed through a soft x-ray neutralizer after size
classification. Upon exiting the neutralizer, the size-dependent fraction of particles that possess at least one positive or negative charge varies from about 41 % for 50 nm particles to 71 % for 200 nm particles (Wiedensohler, 1988). The flow rate through the reactor was kept at 3.5 L min$^{-1}$.

        Particles were directed through the reactor or through a 150 cm L × 0.95 cm OD copper tube bypass, with sampling alternated between the two through the use of an automated 3-way valve. The particle transmission efficiency was calculated
from the ratio of the particle concentrations measured at the outlets of the reactor and bypass using a CPC (TSI Inc. Model 3760A). After a set of initial tests, the static charge on the PFA, PTFE, and ePTFE surfaces was minimized by pushing concentrated bipolar ions generated with an electronic ionizer (Simco-Ion Inc., Fusion) through and around the flow tube for more than 12 hours. Additional measurements of 50 and 100 nm particles were made after minimizing the static charge. The measurements were repeated two or three times for each particle size, with agreement between measurements found to be to
within ± 5 % when sampling the same diameter.

        Gas losses were determined by injecting 10 s pulses of $CO_2$ and $SO_2$ and the transmission efficiencies were determined by measuring the ratio of the concentrations downstream and upstream of the reactor with the $CO_2$ and $SO_2$ analyzers identified above.





## 2.6 UV intensity profile and OH exposure level

The 254 nm intensity at multiple positions inside the reactor was examined using a spectroradiometer (OceanView, Model USB4000 UV-FL) via a fiber optic cable. The influence of the reflective material(s) wrapped around the flow tube was assessed by measuring when it was wrapped only with aluminum-coated Mylar and when it was wrapped with a combination of ePTFE gasket (inner layer) and Mylar (outer layer). The OH production rate and corresponding equivalent exposure was varied by changing the UV intensity, RH, and injected $O_3$ concentration. Here, $OH_{exp}$ is defined as the OH concentration

(molec. $cm^{-3}$) multiplied by the mean residence time of the sample in the reactor. The UV intensity from the lamp was controlled over a range of 50 to 100 % using a lamp manager (BHK. Inc, PN IM10003) by stepping the control voltage from 0 to 5 V. The $O_3$ concentration in the reactor was varied by adjusting the position of a sleeve over the lamp in the ozone generator. To quantify $OH_{exp}$, $SO_2$ was injected with initial mixing ratios ranging from 150 to 250 ppb. For each test, the UV lamp was initially off, and was turned on only after the $SO_2$ concentration measured at the outlet was stable. After the lamp

was turned on, the concentration of $SO_2$ was monitored at the reactor outlet. The typical concentration pattern observed is shown in Fig. 3. OH exposure was quantified for each UV lamp intensity and $O_3$ concentration combination using Eqs. (1) and (2) (Davis et al., 1979; Atkinson et al., 2004). The procedure was repeated two or three times at each UV intensity.

$$d\ [SO_2]/dt = -k_{OH\text{-}SO2}[OH][SO_2] \hspace{3cm} (1)$$

$$OH_{exp} = k_{OH\text{-}SO2}^{-1} \times ln\ [SO_2]_0/\ [SO_2]_f \hspace{3cm} (2)$$

Where:

$k_{OH\text{-}SO2}$          — $9\times10^{-13}\ cm^3\ molec.^{-1}\ s^{-1}$

$[SO_2]_0\ and\ [SO_2]_f$          — $SO_2$ concentrations measured at the reactor outlet without and with the UV lamp
                    turned on



# 3 Results and discussions

## 3.1 Flow dynamic characterization

The 2-D geometry velocity profile simulation result is shown in Fig. 4. The simulation used the actual design and dimensions of the PFA reactor. The flow at the entrance to the sample outlet tube of the main body is assumed to be fully developed and laminar, while an atmospheric pressure boundary condition at the annular outflow boundary and the no-slip condition at all the other boundaries were applied. Though high velocity extends into the central tube flow region above the inlet, within 15 cm from the entrance of the diffuser cone the velocity profile is nearly parabolic, with a decrease in the maximum velocity over the entrance length from 12 cm s$^{-1}$ to 3 cm s$^{-1}$. The simulation suggests that jetting is minimal and that the area influenced by recirculation is negligible.

## 3.2 UV intensity distribution and OH$_{exp}$ level

The normalized UV intensity as a function of distance from the lamp located at the top of the reactor is shown in Fig. 5 (a). The normalized UV intensity is calculated as the intensity at a specified position divided by the maximum measured inside the PFA tube. As expected, an intensity gradient exists, with decreasing intensity with distance from the lamp. The gradient is much steeper when the flow tube is not wrapped with the ePTFE gasket. Without the ePTFE gasket, the intensity near the bottom of the tube is only 15 % of that at the top. Adding the ePTFE resulted in an intensity 30 cm from the bottom that was approximately five times higher than that with only the Mylar. The relative UV intensity enhancement (ER$_{Intensity}$) is shown as a function of position in Fig. 5 (b). The UV intensity is enhanced by a factor of between about 2 and 6 with the addition of the ePTFE layer. In addition to increasing the average UV intensity, the use of the reflective gasket reduced the gradient in intensity, resulting in more uniform OH generation throughout the reactor.

The OH concentration and resulting OH$_{exp}$ were varied by varying the UV intensity, the added O$_3$ concentration, and the RH. Figure 6 (a) shows the sensitivity of OH exposure to ozone concentration and UV intensity. Without the ePTFE wrap around the reactor, the OH concentration ranged from approximately $1.3 \times 10^8$ to $2.9 \times 10^9$ molec. cm$^{-3}$. The corresponding OH$_{exp}$ ranges from $2 \times 10^{10}$ to $4.3 \times 10^{11}$ molec. cm$^{-3}$ s, which is approximately equivalent to 0.15 to 3.3 days of atmospheric exposure based on the reference average OH concentration of $1.5 \times 10^6$ molec. cm$^{-3}$. The effect of the ePTFE wrap on OH$_{exp}$ is shown in Fig. 6 (b). The increased reflectance and UV intensity resulted in a maximum OH$_{exp}$ of approximately $1.1 \times 10^{12}$ molec. cm$^{-3}$ s, equivalent to 8.5 days of atmospheric OH exposure, for the same RH (40 %) and O$_3$ mixing ratio (3.3 ppm) that resulted in an equivalent 3.3 days without the ePTFE. Overall, the highly reflective (and non-absorbing) materials used result in OH exposure comparable to that in other OFRs despite the use of a relatively low power output lamp.

## 3.3 Gas and particle transmission efficiency

Figure 7 shows the transmission efficiency of AS particles with mobility diameter ranging from 50 to 200 nm. As stated above, particle transmission efficiency is calculated as the ratio of the concentration exiting the reactor to that exiting a





copper tube bypass. Concentrations measured upstream and downstream of the copper tube agreed within ±1 %, confirming
minimal loss in the bypass line. We performed two sets of tests: first, following the removal of static charge on the inner
surface of the reactor tube (preliminary removal process), and second, following the additional removal of static charge
between the ePTFE/Mylar wrap and the outer surface of the reactor tube (secondary removal process). The particle
transmission efficiency after removal of only the charge on the inner surface of the tube was 0.39, 0.75 and 0.93 for 50 nm, 80

nm, and 100 nm diameter particles, respectively. With the removal of the static charge on the outer surface of the tube, the
transmission efficiency of 50 nm and 80 nm particles increased from 0.39 to 0.75 and from 0.75 to 0.84, respectively. Each
experiment was repeated twice, with agreement within ±10 % when sampling the same particle size and with the same flow
rate. These results indicate that loss of small particles in the reactor can be significantly reduced by minimizing the static
charge on both the inner and outer surfaces of the reactor tube. The similarity in the resulting 36 % increase in transmission

efficiency of 50 nm particles and the 41 % of those 50 nm particles that are expected to be charged (Wiedensohler, 1988),
suggests electrostatic loss was minimal after the static charge was minimized. Comparison with the particle transmission
efficiency of other types of flow tube reactors with non-conductive wall materials is included in Fig. 7. The PAM reactor
referenced is the horizontal 46 cm L × 22 cm ID glass cylindrical chamber with a volume of 15 L that was described by Lambe
et al. (2011), hereafter referred to as the quartz-PAM. The results show that the particle transmission efficiency through the

PFA reactor is higher than that for the quartz-PAM and TPOT reactors, with that for 50 nm particles being 0.75, 0.18, and 0.40
for the PFA, quartz-PAM, and TPOT, respectively.

The experimental configuration used to measure the loss of $SO_2$ and $CO_2$ is similar to that used to characterize the gas
RTD. The penetration efficiencies of $CO_2$ and $SO_2$ were 0.90 ± 0.02 and 0.76 ± 0.04, respectively. The wall loss for most
precursor species is expected to be equal to or less than the 24 % found for $SO_2$ because it is a good surrogate for wall-adhering

species (Lambe et al., 2011; Ahlberg et al., 2017; Huang et al., 2017). For comparison, Lambe et al. (2011) reported that the
measured $CO_2$ and $SO_2$ transmission efficiencies for the TPOT were 0.97 ± 0.10 and 0.45 ± 0.13, respectively, and for the
quartz-PAM were 0.91 ± 0.09 and 1.2 ± 0.4, respectively.

The fate of low-volatility organic compounds (LVOC) that can condense onto particles, stick to the reactor walls,
react with OH, or exit the reactor before condensing can be evaluated using the approach described by Palm et al. (2016). Based

on the simple model they present, LVOC wall losses for the PFA flow tube have an upper limit of approximately 30 % for a
residence time of 130 s, which is comparable to that observed for $SO_2$ (24 %). Although the LVOC fate method is strongly
dependent on the design and the geometry of the reactor, the consistency between the estimated loss and that measured for
$SO_2$ suggests the value is a reasonable estimate of the vapor loss for our design. Losses of some gases are expected to be greater
in this OFR than in most others because of its larger surface area to volume (A/V) ratio of 0.53 cm$^{-1}$, which is greater than that

of the PAM reactor, while the mean residence times of the two are similar. However, losses of some gases may be lower as
well because only the central core flow is subsampled, all Teflon materials are used, and, as is described in the next section,
the RTD is comparatively narrow, which suggests less mixing than in other OFRs.





## 3.4 Gas and particle residence time distributions

The residence time probability distribution functions for particles and gases are shown in Fig. 8 (a) and (b). Reporting the results as normalized distribution functions facilitates comparison of the flow characteristics of reactors of different shapes and sizes. RTDs of idealized devices and those reported for CPOT and quartz-PAM are also shown in Fig. 8 (a) and (b) for comparison (PAMWiki, 2019). The residence time probability distribution function is defined as the normalized measured concentration ($C_{out}(t)$) divided by the total area of the normalized pulse (Fogler, 2006; Simonen et al., 2017), as described in Eq. (3) below. The average residence time was calculated as the summation of the product of the measured concentration and

the corresponding residence time, all divided by the total area of the pulse.

$$PDF(t) = \frac{C_{out}(t)}{\int_0^\infty C_{out}(t)dt}$$

(3)

     The residence time distributions of particles and gases in the PFA reactor shown in Fig. 8 (a) and (b) approach those expected for laminar developed flow. Measured RTDs for both particles and gases have relatively short tails at longer times

compared with the ideal laminar flow pulse, as is expected because only the center ~50 % of the sample flow is subsampled and directed to the analyzers. Relative to the total flow through the reactor, the subsampled core has a narrower velocity range and less interaction with the walls. Extraction of the side purge flow may also help by preventing recirculation near the outlet. The RTDs measured with the UV lamp turned on are only slightly broader than those with it turned off. Previous studies report that UV lamps broaden the RTD because they heat the reactor walls and enhance convection inside the reactor (Simonen

et al., 2017). Significant degradation is not observed in the PFA reactor, presumably because of the use of a comparatively low-power light source, circulation of air through the reactor housing, and the reactor being oriented vertically with the lamp at the top to promote stratification and to minimize convective mixing. Reversible uptake by the walls is responsible for the broader RTD for $SO_2$ relative to that for $CO_2$.

     We also measured the RTD for AS particles with the side purge flow turned off. This condition was numerically

simulated in COMSOL 5.4 by coupling the Laminar Flow and the Transport in Dilute Species packages. The results are compared with the RTD of that obtained experimentally in Fig. 9. The experimental and simulation results are very similar with only a slightly steeper slope in the simulation curve. The experimental RTD measured with the purge flow turned on and only the core flow subsampled is also included in the figure to clearly show the improved response.

## 3.5 SOA yield measurements

Secondary organic aerosol yields (Y) are defined as the mass of OA formed ($\Delta C_{OA}$) per reacted precursor mass ($\Delta HC$) (Odum et al., 1996). The measured yields of *m*-xylene and α-pinene SOA as a function of OH exposure and organic aerosol concentration ($C_{OA}$) are shown in Fig. 10 (a) and (b). Here, the SOA yields are corrected for size-dependent gas and particle





losses, with an average magnitude of the combined correction of 25 %. For comparison, the magnitude of the particle wall loss correction of the PAM reactor was 32 % ± 15 % (Lambe et al., 2015). The $C_{OA}$ was calculated by multiplying the volume

concentration measured with an SMPS by an assumed density of 1.2 g cm$^{-3}$. The mixing ratios of *m*-xylene and α-pinene introduced into the PFA reactor were in the ranges of 20-118 ppb and 13-145 ppb, respectively. The OH$_{exp}$ was not measured simultaneously during the yield experiments but was estimated as a function of O$_3$, RH, and UV lamp power from the results described in Sect. 3.2 and the assumption that the OH reactivity was the same for both sets of measurements. As expected, the SOA yield was observed to be dependent on OH exposure and aerosol mass concentration. The *m*-xylene SOA yield was

0.22 at $3 \times 10^{11}$ molec. cm$^{-3}$ s OH exposure and an OA mass concentration of 46 µg m$^{-3}$ and the α-pinene SOA yield was 0.37 at $3 \times 10^{11}$ molec. cm$^{-3}$ s OH exposure and a mass concentration of 178 µg m$^{-3}$.

The measured yields are compared with those reported by Lambe et al. (2011) for the TPOT (for 262–263 ppb precursor mixing ratio), the quartz-PAM (78–88 ppb), and the Caltech environmental chamber (14–48 ppb), and by Ahlberg et al. (2017) for the aluminum PAM (14-179 ppb of α-pinene and 43-395 ppb of *m*-xylene). The comparisons as a function of

$C_{OA}$ are shown in Fig. 11 (a) and (b). The SOA yields are higher in the PFA reactor than those in the quartz-PAM and TPOT but lower than in the aluminum PAM. The α-pinene SOA yields in the PFA reactor (0.37 ± 0.02) and Caltech chamber (0.42 ± 0.06) agreed within 12 % for comparable OH exposures ($\sim 10^{11}$ molec. cm$^{-3}$ s). A contributor to differences in yield among the OFRs is variation in OH$_{exp}$, which, as noted above, was not measured during the yield experiments. Our estimates of OH$_{exp}$ neglect the impact of varying OH reactivity (OHR), which is defined as the summation of the product of the concentrations of

species that react with OH and their reaction rate constants (Li et al., 2015; Peng et al., 2015). During our experiments, the maximum OH reactivities for the *m*-xylene and α-pinene experiments were 34 s$^{-1}$ and 103 s$^{-1}$, respectively, which is higher than the 5.5 s$^{-1}$ estimated for the SO$_2$ experiments that were used to determine the dependence of OH$_{exp}$ on RH, O$_3$ concentration, and lamp power. This is also a source of uncertainty in PAM yields that were reported in Lambe et al. (2011) and is estimated by Li et al. (2015) to result in a factor of 2 uncertainty in OH$_{exp}$ obtained from their model-derived equation.

Another contributor to differences between the yields measured with the PFA reactor and those with the other OFRs considered is the difference in operation mode, with the PFA reactor operated in OFR254 mode and the PAMs and TPOT in OFR185 mode. Differences in O$_3$ concentrations and resulting partitioning between reaction with O$_3$ and OH are expected to be more important for α-pinene than for *m*-xylene. The formed SOA is dependent on the reactivity of one or more of the SOA-forming compounds and the oxidant concentrations (McFiggans et al., 2019). For the same O$_3$ mixing ratio (3.3 ppm) and OH exposure

($3 \times 10^{11}$ molec. cm$^{-3}$ s) described above, the reactivities of α-pinene towards O$_3$ and OH are estimated to be $6.8 \times 10^{-3}$ s$^{-1}$ and $111 \times 10^{-3}$ s$^{-1}$, while that of *m*-xylene towards OH is estimated to be $50 \times 10^{-3}$ s$^{-1}$.

### 3.6 Seed particle SOA enhancement

The influence of seed particle concentration was investigated by measuring SOA yield for varying ratios of the mass concentrations of α-pinene and AS seed. For all experiments a constant flow rate (0.7 L min$^{-1}$) containing the AS seed particles

was introduced together with a varying mixing ratio of α-pinene (8-30 ppb). Using the same method that was presented in





Sect. 2.5, a DMA generated a narrow mode of AS seed particles centered at a diameter of 200 nm. The average mass concentration of the AS aerosol throughout the experiments was 40 µg m$^{-3}$. The $O_3$ concentration, RH, and UV lamp power were the same for all measurements, with a resulting $OH_{exp}$ of about $2 \times 10^{11}$ molec. cm$^{-3}$ s. Measurements for each precursor concentration were repeated two or three times, with agreement between measurements to within $\pm$ 10 %. Figure 12 (a) and
(b) show the volume size distributions for one set of experiments with and without added AS particles. The results show that the addition of seed particles suppresses the nucleation mode as condensation on the larger particles is favored. The concentration of α-pinene SOA increased with the addition of high concentrations of seed particles, as is expected because the increased surface area promotes condensation on the aerosol and decreases the fraction of low volatility oxidation products that reach and are lost to the walls or are further oxidized in the gas phase. In these experiments, the yield increased by as
much as a factor of 3 at the minimum precursor:seed mass ratio of about 2. The magnitude of the enhancement decreased with increasing precursor:seed ratio and was within the run-to-run variability for ratios exceeding about 5.

### 3.7 Aerosol formed from oxidation of ambient air

Ambient air from outside our lab at the UCR College of Engineering - Center for Environmental Research and Technology (CE-CERT) in Riverside, CA was processed by the PFA reactor for several days in January, 2020. Figure 13
(a) and (b) show results for a 30-hour period (Jan. 7-8, 2020) and a 6-hour period on Jan. 8, 2020. Throughout the sampling period, the SMPS alternated through sets of three measurements of the processed aerosol at the exit of the reactor and sets of two measurements of unprocessed aerosol that bypassed the reactor through a copper tube. Each cycle of five measurements lasted 21 min. The OHR during the measurement period is estimated to be 5-30 s$^{-1}$ (Steiner et al., 2008; Mao et al., 2012).

Time series of aerosol mass concentrations calculated from integration of the SMPS size distributions are shown in Fig. 14 (a). The mass concentration of the aerosol exiting the reactor was corrected for the fractional dilution by the injected $O_3$ flow and for size-dependent gas and particle transmission efficiencies. The aerosol mass concentration increased significantly in the reactor during the oxidation process. A relative SA enhancement ($ER_{SA}$) is defined here as the ratio of the mass concentration of SA divided by that of the ambient (unprocessed) aerosol, with the SA simply defined as the difference
between the processed and unprocessed aerosols. The $ER_{SA}$ for the same sampling period is shown in Fig. 14 (b). A consistent diurnal pattern was not observed throughout the sampling period. The SA mass concentration was an average of 1.8 times that of the ambient aerosol during the selected period. More SA formation was observed during nighttime on Jan 8, while decreasing amounts formed until around noon. The maximum enhancement due to SA formation was observed in the late afternoon on Jan. 7, when the SA mass concentration was approximately 7 times that of the ambient aerosol. A small SA enhancement was
also observed during the late afternoon on Jan. 8. The overall temporal pattern likely reflects the impact of traffic related emissions from nearby roads, including a major highway that is about 1.5 km away. In the future there is a need to add more comprehensive measurements of the chemical composition of the particulate and gaseous species.



## 4 Summary

A new all-Teflon reactor, the Particle Formation Accelerator (PFA), was designed, constructed, and characterized
using both experimental measurements and CFD modeling. Its performance was examined and evaluated through laboratory
measurements and with ambient air. The reactor response and characteristics were compared with those from a smog chamber
(Caltech) and other oxidation flow reactors (the Toronto Photo-Oxidation Tube (TPOT), Caltech Photooxidation Flow Tube
(CPOT), and quartz and aluminum versions of Potential Aerosol Mass reactors (PAMs)). Our results show that $OH_{exp}$ can be
varied over a range comparable to that of other OFRs, with the dependence on UV lamp power, RH, and $O_3$ concentration
characterized and reported. The particle transmission efficiency is over 75 % in the size range from 50 to 200 nm after
minimizing static charge on the PFA, PTFE, and ePTFE surfaces. The gas transmission efficiencies of $CO_2$ and $SO_2$ are 0.90
$\pm 0.02$ and $0.76 \pm 0.04$, respectively, with the latter comparable to estimated transmission of LVOCs through the PAM reactor.
Computational simulation and experimental verification of particle and gas residence time distributions (RTDs) show that the
flow through the reactor is nearly laminar, with narrower RTDs than reported for OFRs with greater diameter-to-length ratios.
The mass yields of SOA from the oxidation of α-pinene and $m$-xylene, and the effect of seed particles on those yields,
were investigated. At comparable OH exposure, the $m$-xylene and α-pinene SOA yields are slightly higher than those in the
quartz-PAM and TPOT, but lower than in the aluminum-PAM. A likely contributor to differences in yields between the PFA
and other OFRs is the uncertainty in $OH_{exp}$, which was not measured simultaneously during the yield measurements and was
determined from separate experiments for which the OH reactivity differed. The α-pinene SOA yields in the PFA reactor (0.37
$\pm 0.02$) and Caltech chamber ($0.42 \pm 0.06$) agree within 12 % for comparable OH exposures ($\sim 10^{11}$ molec. $cm^{-3}$ s). The presence
and concentration of seed particles was shown to have a significant effect on SOA yield. At a nominally fixed OH exposure
of $2 \times 10^{11}$ molec. $cm^{-3}$ s, the α-pinene SOA yield for the minimum precursor:seed mass ratio of about 2 was about 3 times
that when no seed particles were added. The magnitude of the enhancement decreased with increasing precursor:seed ratio and
was within the run-to-run variability for ratios exceeding about 5. The SA production from ambient air was studied in
Riverside, CA. The mass concentration of SA formed in the reactor was about twice the mass concentration of the ambient
aerosol at the same time.

Overall, the computational and experimental results indicate that the PFA reactor is suitable for laboratory studies
and for field use that includes measurement of rapidly changing ambient concentrations. Future efforts will include adding
direct measurement of $OH_{exp}$ during measurements, development of an $OH_{exp}$ estimation description for the PFA reactor
comparable to that reported for other OFRs, and further exploring the influence of OH reactivity on $OH_{exp}$ and of seed particles
on SOA yield. We will also expand upon measurements of the composition of the particulate products and gaseous precursors
during one or more field studies to evaluate how well the PFA reactor simulates atmospheric chemistry that typically requires
hours or days.

*Data availability.* Data presented in this work are available from the authors.



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





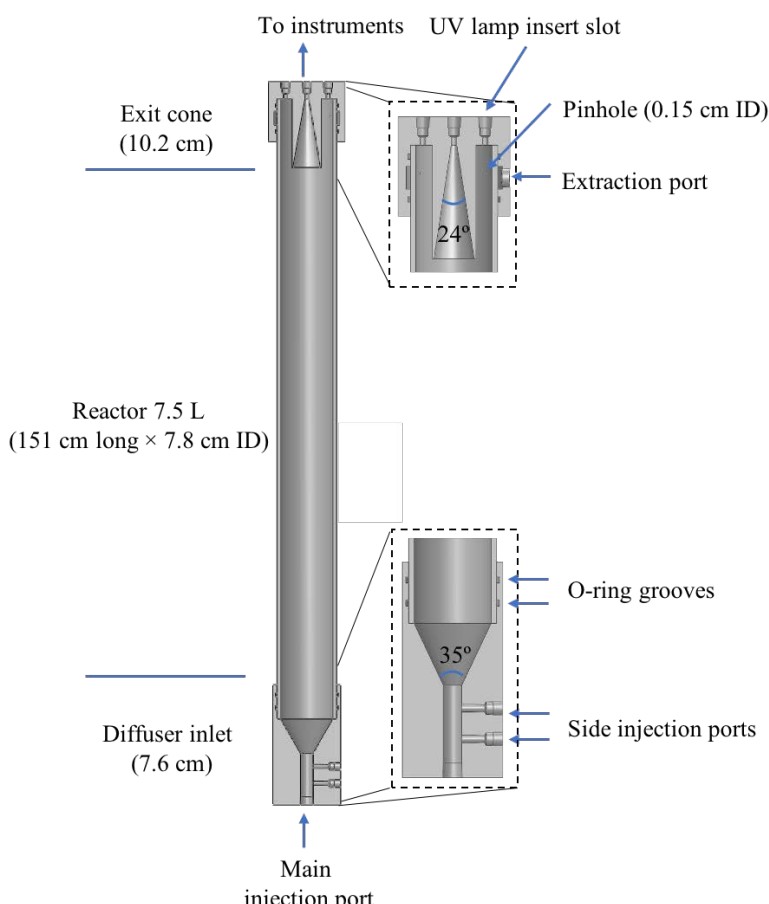


**Figure 1: Cutaway view of the PFA reactor.**





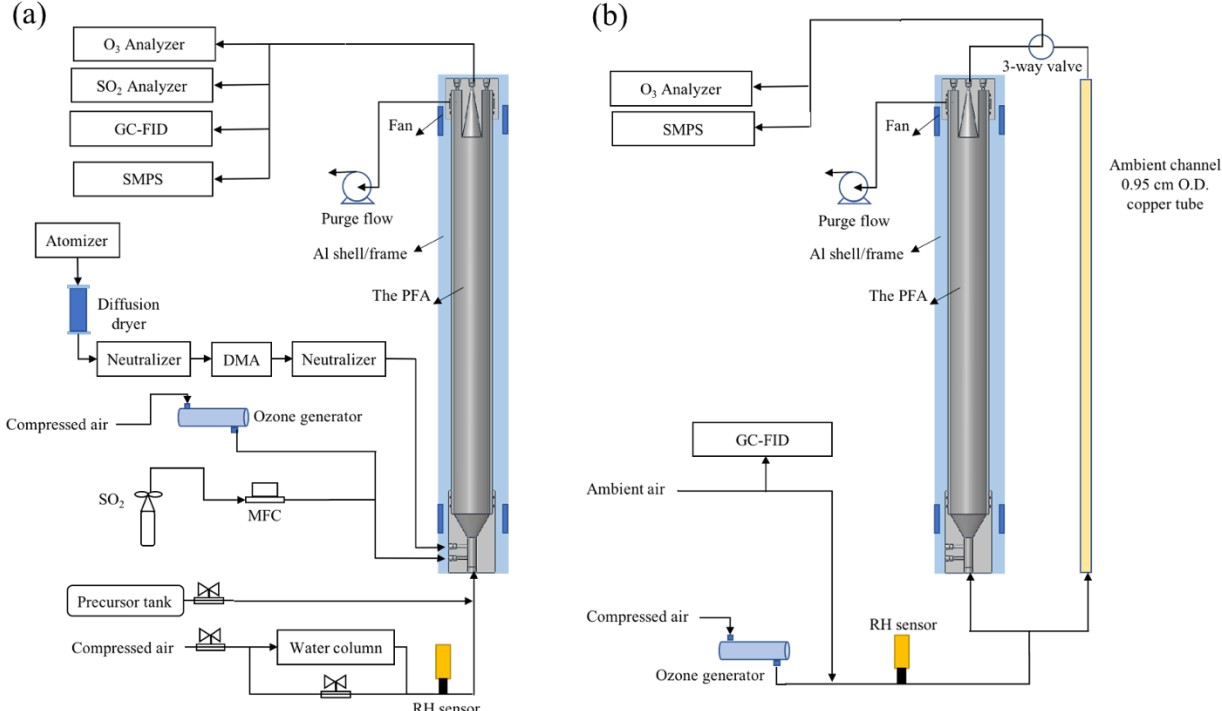

**Figure 2: Schematic diagram of the PFA and associated experimental setup for the laboratory and (b) field experiments.**






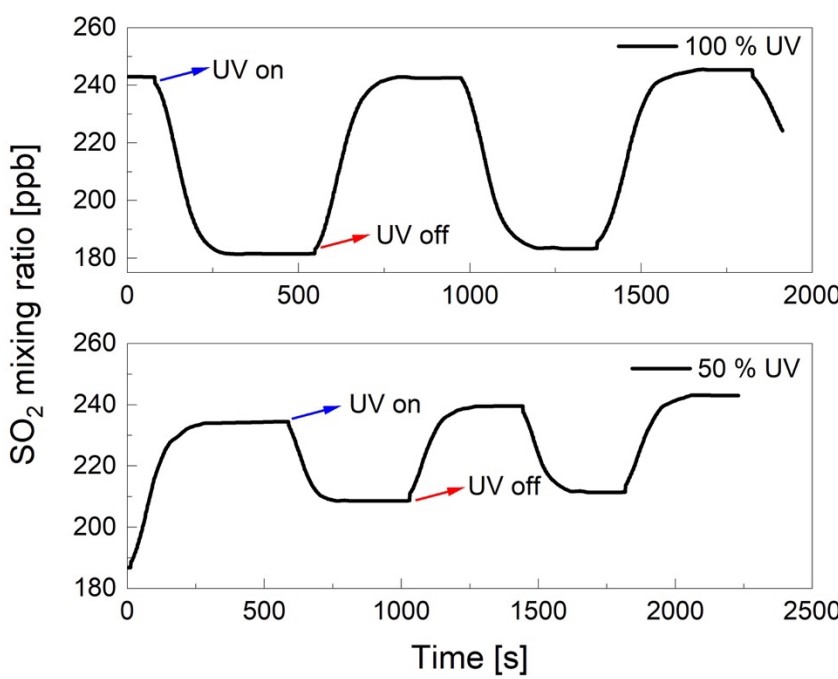

**Figure 3: Example results from experiments to characterize OH exposure using injected SO₂.**




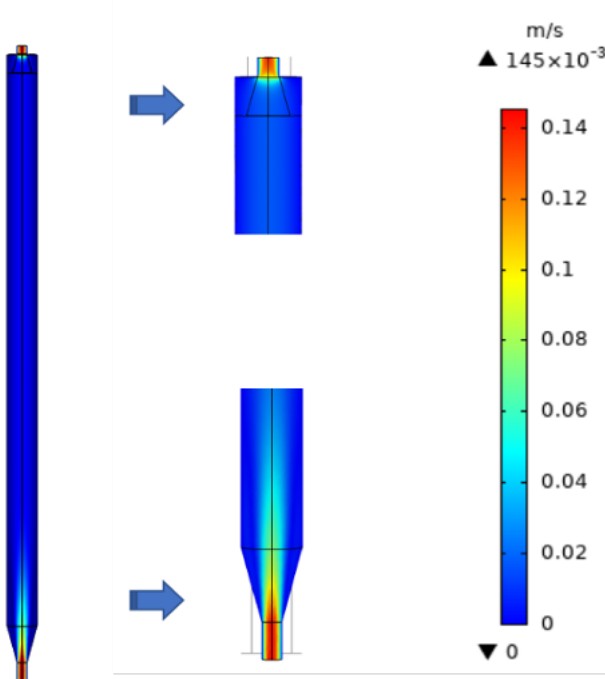

**Figure 4: CFD simulation results of the velocity distribution in the PFA reactor.**






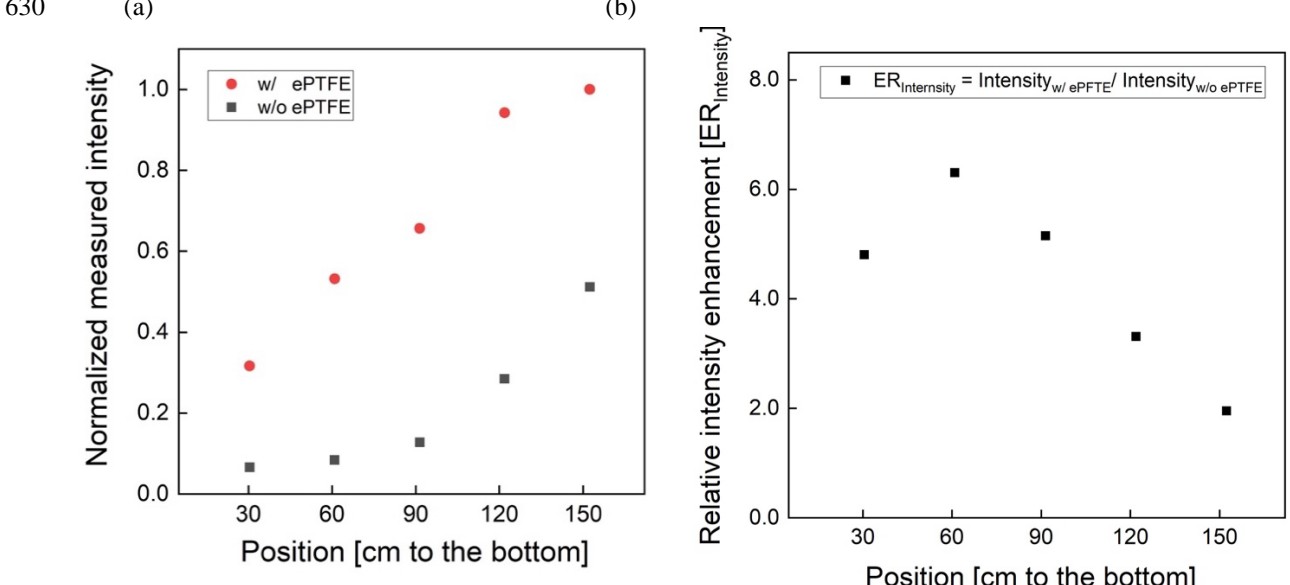

**Figure 5: Relative UV intensity profile (a) and intensity enhancement (b) achieved when the flow tube was wrapped with reflective ePTFE gasket.**






(a)

(b)

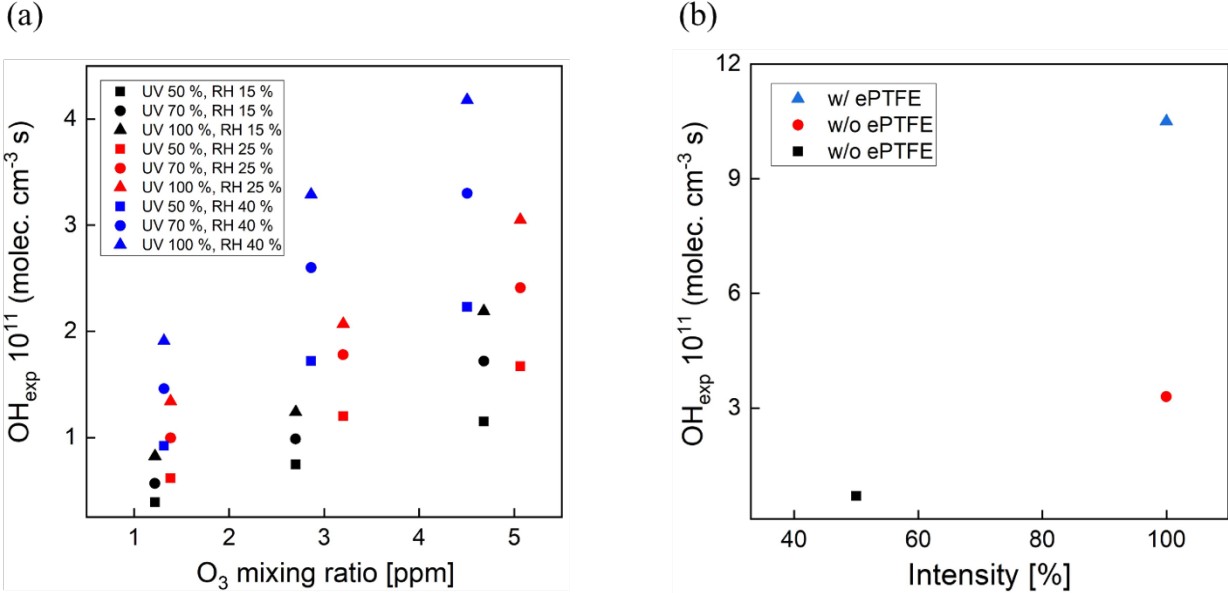

**Figure 6: Variations in the concentration of OH (a) as a function of the O₃ mixing ratio (1.3-5.5 ppm) with varying UV lamp power (50 %, 70 %, 100 %) and RH (15 %, 25 %, 40 %) and (b) as a function of the UV intensity with and without ePTFE wrapped around** 640 **the flow tube.**



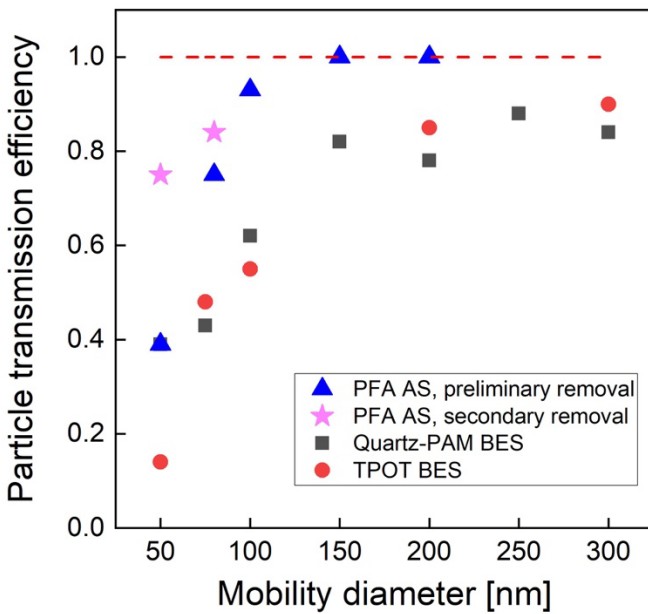

**Figure 7: Measured particle transmission efficiency of the PFA, quartz-PAM, and TPOT flow reactors as a function of mobility**
**diameter for bis(2-ethylhexyl) sebacate (BES; circles and squares) and ammonium sulfate (AS; triangles).**





(a)  (b)

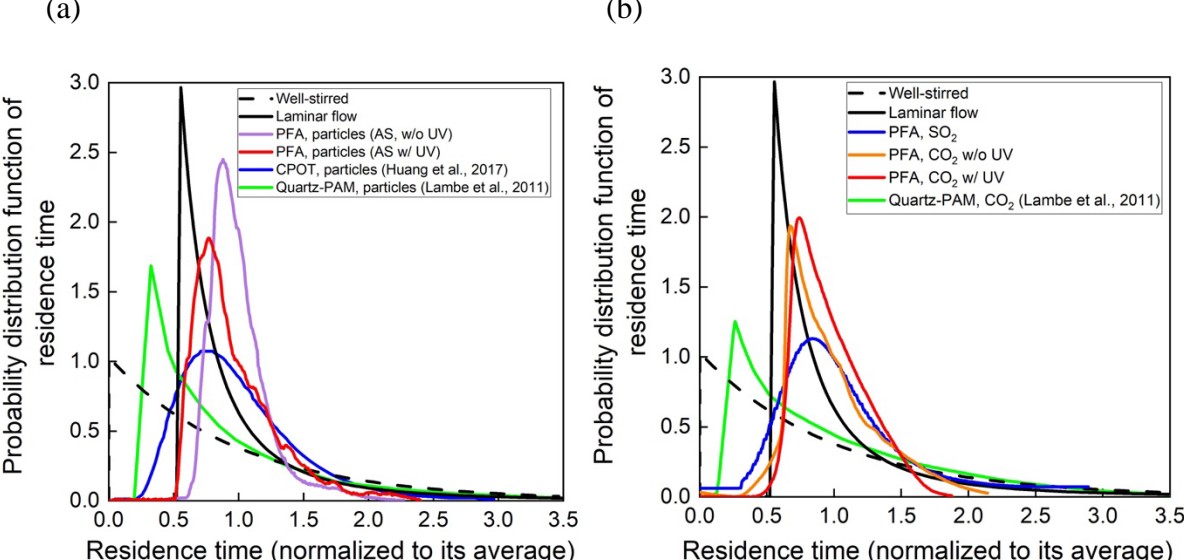

**Figure 8: Residence time probability distribution functions of the PFA, CPOT, and quartz-PAM flow tubes as a function of residence time for (a) particles and (b) gases.**





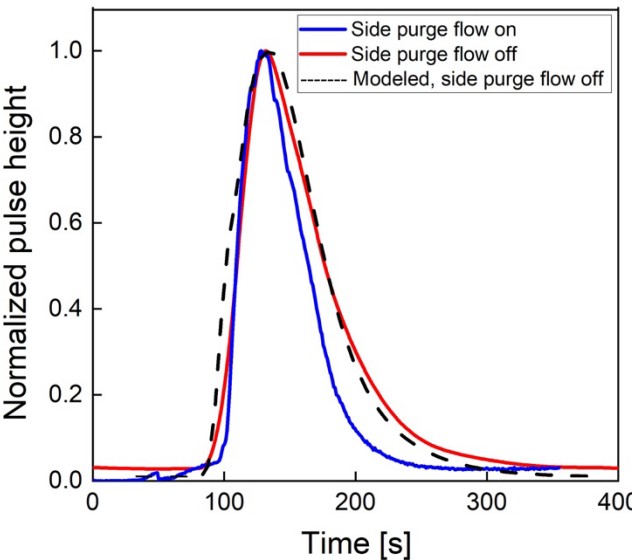

**Figure 9: Measured response of pulse injection of AS particles with the side purge flow on (blue solid line) and off (red solid line), and the COMSOL simulation of the configuration with it turned off (black dashed line).**






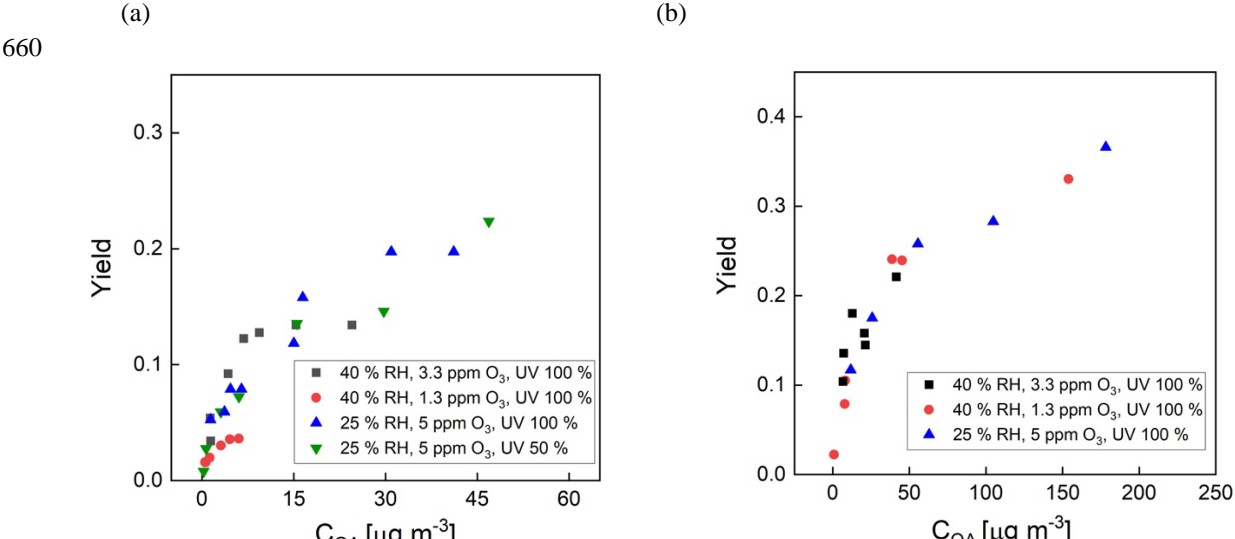

**Figure 10: SOA yield as a function of organic aerosol concentration ($C_{OA}$) for (a) *m*-xylene SOA and (b) α-pinene SOA generated in the PFA reactor. Marker color reflects experimental combinations of UV intensity, $O_3$ mixing ratio, and RH.**




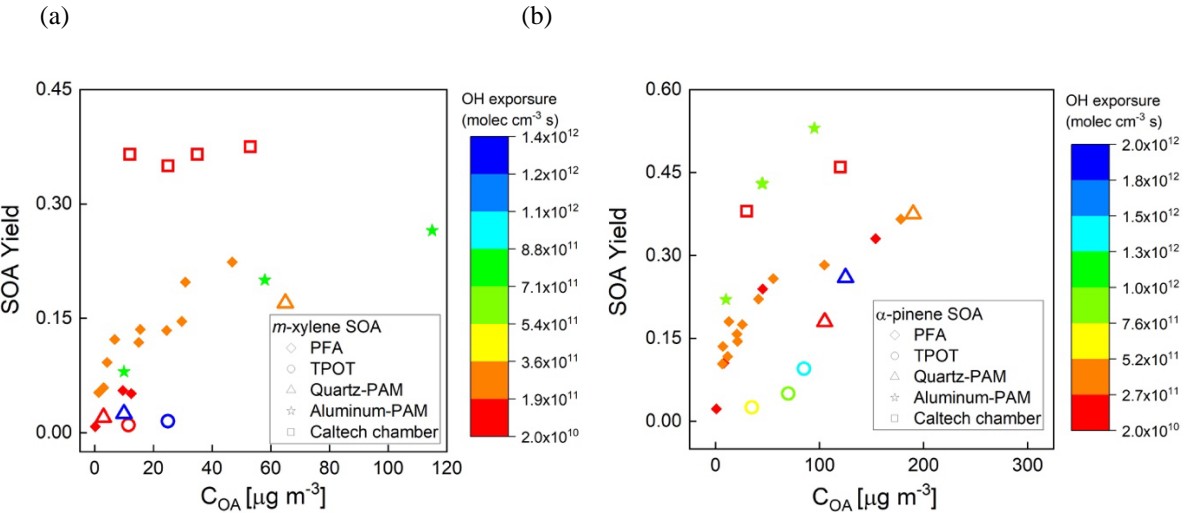

**Figure 11: Comparison of SOA yields as a function of organic aerosol concentration (C$_{OA}$) with those reported for other OFRs and one large Teflon chamber. (a) *m*-xylene SOA and (b) α-pinene SOA.**





(a)                                                      (b)


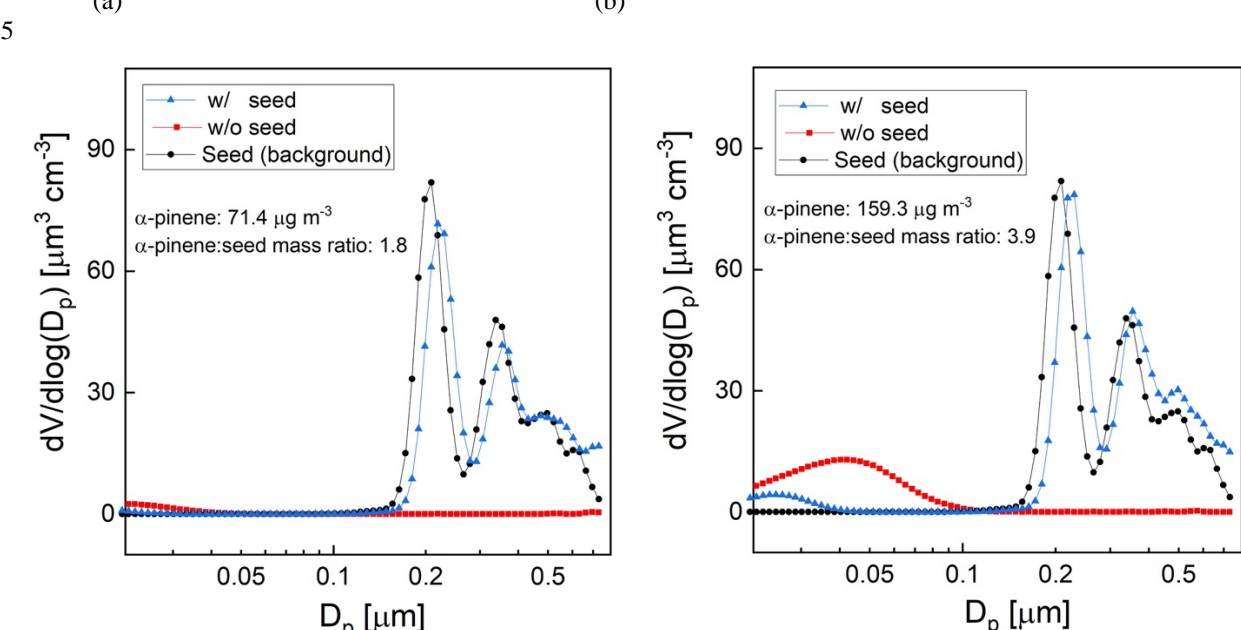

**Figure 12: Example sets of volume size distributions from experiments evaluating the impact of adding AS seed particles on SOA yield. The precursor:seed mass ratio is (a) 1.8 (b) 3.9.**




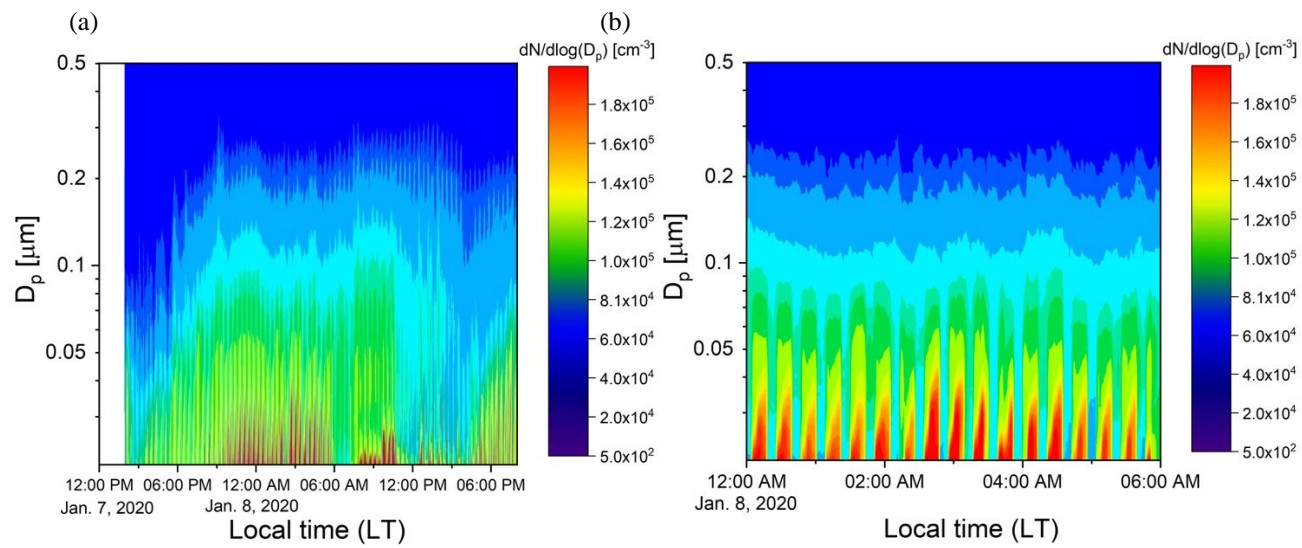

**Figure 13: Example time series of size distributions of the aerosol processed by the PFA reactor and that which bypassed it over (a) 30 hours on Jan. 7-8, 2020 and (b) 6 hours on Jan. 8, 2020. The bands of high concentration were measured when the aerosol and ambient air were processed through the reactor.**






(a)                                                    (b)

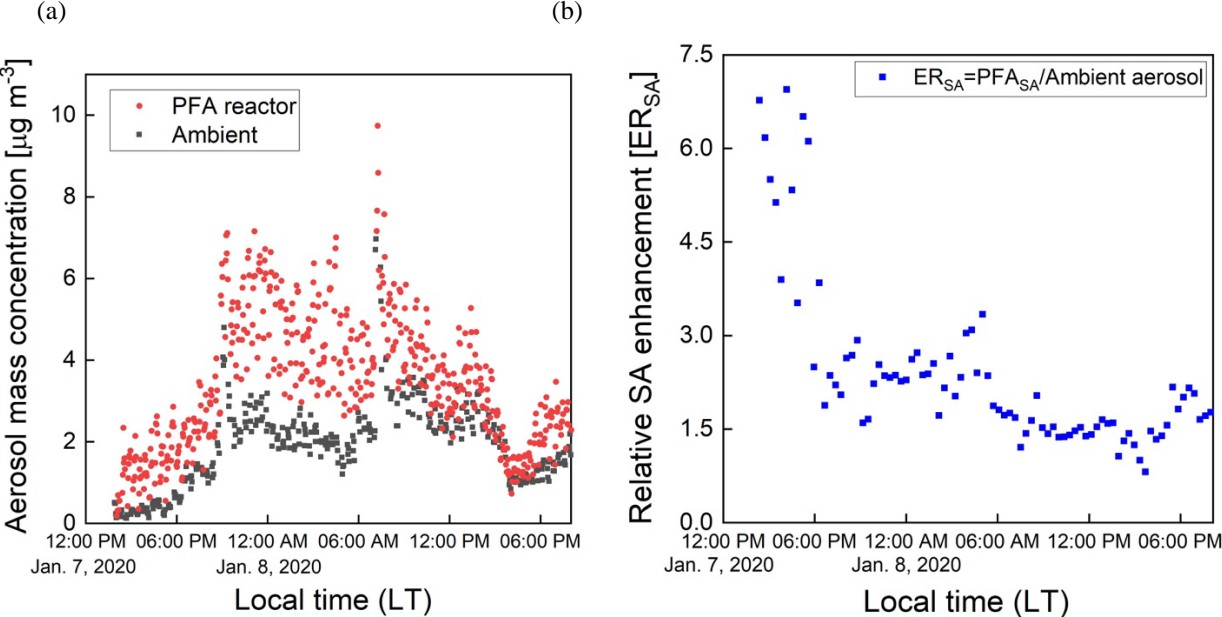

**Figure 14: Time series of mass concentrations of the aerosol exiting the PFA reactor and that bypassing it (a), and (b) the relative enhancement of the mass concentration due to SA formation.**