# Peer review of "Design and characterization of a new OFR for laboratory and longterm ambient studies"

_Atmospheric Measurement Techniques, 2020_

## Referee Comment (RC1) · Anonymous Referee #1 · 24 Nov 2020

Reviewer comments on manuscript amt-2020-373, "Design and Characterization of a new OFR: The Particle Formation Accelerator (PFA)"

The authors present here a description and investigation of a new oxidation flow reactor that is made entirely from Teflon. A variety of OFRs have been developed over the past decade from a range of materials, but few have been previously designed from Teflon despite its high inertness and low vapor wall losses. The specific geometry and design of each has important implications for interpreting the data they produce. It is consequently valuable and important to conduct (and submit for peer-review) the detailed sorts of testing described here, and this is a fitting journal for this work. This manuscript

presents a detailed description of the design and examination of its residence times, wall losses, and results, and it should be published. My concern, however, is not so much with the science (though there are some specific comments below), but in the presentation. Overall, I think many of the Figures could be made more clear, and many of them (and some of their discussion) does not need to be incldued in the main body and could be moved to an SI.

General comment:

(1) In the descriptions of the operation of the PFA, the authors spend a lot of time discussed operating modes that are not necessarily pertinent to the final operational approach. For instance, in some figures they show the impacts of operating with the side purge flow off, or without the reflective ePTFE. These feel to me like advancements the authors figured out along the way that don't describe the actual recommended operation of the system (i.e., the authors do not seem to recommend ever operating with the side purge flow off). Consequently, they seem better suited to a brief mention in the main text and deeper discussion in a Supplementary Information for the more interested reader. The discussion of the static removal is similar, though I think may warrant a discussion in the main text as the static losses might be expected to be the main issue with a non-conductive OFR. Figures (and associated discussions )that I think are more detail than are maybe relevant to the main text are: Figure 3 - This figure is illustrative of the detailed description of the text, I don't think it really adds much Figure 5 - shows the importance of the the reflective sheath, but since the reflective sheath is used throughout the manuscript and is "part" of the instrument, this mostly shows work done along the way, and is not of central importance to the rest of the work. One option would be to show the gradient by pairing Fig. 5a and Fig 6a as one figure, as Fig 5b (which is just the ratios shown in 5a) and 6b (which doesn't have much information) don't seem critical. Figure 9 - again, as with Figure 5, this mostly shows work done along the way, and is not really germane to the final instrument. The interested reader might want to dig into this, but since the average reader is probably

more interested in the final configuration, this could go to the SI

This is a general issue, not just related to these specific figures. Overall, I think discussions of some of the methods are perhaps a bit too detailed, and then interpretation of the demonstration and scientific data at the end is a bit sparse.

Specific comments:

(2) Line 89-90: I am not sure, but I thought there was also a teflon film OFR in use by the Jimenez group at some point? Bill Brune also published one in 2007: https://acp.copernicus.org/articles/7/5727/2007/acp-7-5727-2007.pdf . Given that one of the big values/advances of the present work is the all-Teflon construction, some discussion of Teflon OFRs should be included.

(3) Line 107: It is confusing to call it the PFA. I get that it is a play on the fact that it is made of PFA, but it muddies the discussion somewhat.

(4) Line 124: In Figure 1 the annulus and pinholes are not really clear, so it took a little bit of re-reading and examining to understand exactly how it is designed and works, and I'm still not totally sure I have it figured out.

(5) Line 280: Not really a 36% increase in transmission efficiency, more accurately a 36 percentage point increase, maybe state as "36% of particles that are no longer lost"

(6) Line 284: Not clear, is the data in Figure 7 directly from the referenced work?

(7) Line 309: I'm not a statistician, so I'm not sure whether or not this is the right metric or approach - why adjust residence times by the average? Why not normalized to the nominal residence time (V/Q)? Isn't that the metric that is used for the laminar flow case? For example, in Figure 8b, $SO_2$ comes out sooner than expected from laminar flow or than $CO_2$, which the authors attribute to reversible uptake - but that should delay the $SO_2$ response, not speed it up. By delaying response, the average RT is shifted later, and the apparent speed of the early eluting $SO_2$ is faster. I think all of this would be easier to interpret (and perhaps more meaningful) if the nominal RT were

used instead of the average.

(8) Lines 342-361: How should the scientific community be dealing with or thinking about these yields? Looking at Figure 11, OFRs seem to be saying that the yield for a-pinene at a give Coa and OHexp could easily vary by a factor of many (e.g. ∼5-50% for Coa = ∼100ug/m3 and OH in the range of 3-10x10ˆ11). This is, of course, not a problem the authors are solely responsible for, but it makes me wonder about the utility of using yield comparisons to validate or understand the OFR. It is reassuring, I suppose, that the PFA is in the middle of all this noise (as opposed to outlying), but are we (the reader and/or the authors) really learning anything from this intercomparison? Is there some way to make sense of this spread other than the broad "uncertainty in OHexp"? Maybe the points in Figure 11 could at least indicate different OFR modes, or something that might explain some of these differences?

(9) Line 366: In Figure 12 it looks like the seed is multi-modal but the test refers to a narrow mode at 200 nm - is this due to multiple charging in the DMA?

(10) Line 374-376: Adding a panel to Figure 12 showing the trend in yield as a function of pinene:AS would aid this discussion

(11) Line 384: Both of these citations refer to specific conditions and locations in the early 2000's. How are the author's using these to estimate the stated OHR?

Figure comments:

(12) Figure 2 is missing "(a)"

(13) Figure 3 labels are confusing because the arrows could refer to "that point onward" or to the region they are coming from. It would be clearer to indicate with colored regions or a binary on/off trace when UV was on or off.

(14) Figure 6 could be made more clear - too much going on in the legend with lots of repetition. Also not clear if 6a is with or without the ePTFE. I gather it is without? Why are only two points from 6a reproduced on 6b? Why not just include the one point w/

ePTFE shown in 6b into 6a and make all the labeling more clear?

(15) Figure 9 - why not also include the modeled with the side flow on?

(16) Figure 10 legends are similarly busy and repetitive requires a lot of looking back and forth. Some estimate of OH exposure on these plots might help.

(17) Figure 11 - make the legend markers darker so they can be seen more easily

---

## Referee Comment (RC2) · Anonymous Referee #3 · 28 Nov 2020

Xu and Collins presented a comprehensive study of a newly designed flow reactor, the particle formation accelerator (PFA), the basic idea of which is from the oxidation flow reactor (OFR). Different from the commercialized OFR, the PFA is made of PFA-Teflon and vertically oriented, thus having different effects on potential losses of both vapor molecules and particles as well as the fluid dynamics inside. The authors characterized the residence time distributions (RTD), transmission efficiencies, yields, and aerosol forming enhancement of the PFA. The manuscript is overall well written and the introduction of the PFA will be of interest to the OFR community. It fits the scope of AMT and should be accepted for publication after considering the following minor revisions:

1. I think in the Introduction part it is worth mentioning that the PFA is vertically oriented and the heating source is on the top, thus minimizing the convection and maintaining laminar flow profile, which is different from other oxidation flow reactors.

2. Section 2.1 is a little difficult to follow and it does take me a while to figure out the actual design. I think it needs to be rearranged with subsections, e.g., inlet, main tube, outlet, lamps, etc., to make it clearer.

3. Line 121: based on the parabolic flow assumption and the dimension (2.1 cm ID of the cone and 7.8 cm ID of the tube), it is just about 14% flow that goes into the cone ideally. That 50% flow is extracted at the cone suggests that it is not the parabolic profile at the outlet. Maybe clarify this point here.

4. About the lamp: if I understand correctly, the lamp is exposed directly to the flow? Also, since Sections 2.6 & 3.2 presented the experimental measurement of light intensity as a function of the distance to the bottom, it is worth mentioning it in Section 2.1 at least or rearranging the two sections.

5. About the position of ePTFE: is it inside the PFA tube or outside the PFA tube but inside the layer of aluminized Mylar?

6. Line 142: it is not very clear how the two U-bolts hold the tube inside the aluminum tube.

7. Line 148: is the temperature referring to the gas coming out of the PFA tube or that coming between the shell?

8. About the experimental setup: how is the gas getting through the tube, by pulling due to vacuum at the outlet or pushing at the inlet? For each case, there should be an excess flow (or open port), but Fig. 2 cannot tell. I guess it is pulled by the vacuum at the outlet?

9. About purge flow: does this flow determine the actual flow fraction that goes into the outlet cone?

10. About the charge effect on particle wall loss: does it have a preference on a specific polarity when there is a static charge on the surface?

11. Line 221: how is the gas transmission efficiency determined by pulse injection? Does the concentration downstream refer to integrated concentration?

12. Section 3.1: same as comment 3, how does the simulated velocity distribution look like at the outlet w/ and w/o purge flow?

13. Figures 5 & 6 show that even with the high-reflective layer, the UV intensity at the inlet can be one-third of that at the outlet and the corresponding OH concentrations could be the same ratio. Thus there could exist a transition regime in the reactor that a compound reacts with $O_3$ and OH equally. Though it does not affect the OH exposure calculation, knowing the distributions of $O_3$ and OH along the tube will be helpful to analyze the competing reaction channels. If this point is out of the scope of this manuscript, it is worth mentioning that point though.

14. For the transmission efficiency of sticky vapor molecules, I am wondering if the transmission efficiency increases as the Teflon wall gets saturated after absorption/adsorption for a long time? The same question goes to Section 3.5, will the SOA yield increase after the experiment runs for a long time when the Teflon wall gets saturated with the sticky vapor molecules? For experiments with seeds, what is the timescale loss to the wall versus that to the particle surfaces?

15. Equation 3: since RTD is the term used in the context, PDF may be replaced with RTD.

16. Line 324: what is the purpose of turning off the purge flow?

17. Lines 351 and 360: it is worth noting here that the reactivities of OH and VOC are different.

18. Line 384: How is the ambient OHR determined?

19. Figure 10: For experiments under the same condition, how are the different data points from since the flow reactor is usually running at a steady-state? If it is because of different injected VOC concentrations, it is better to mention that in the caption.

---

## Referee Comment (RC3) · Anonymous Referee #2 · 23 Dec 2020

Xu and Collins present the evaluation of a custom PFA OFR design by performing characterization studies that included measurements of residence time distributions, gas and particle transmission efficiencies, yields of laboratory SOA generated from OH oxidation of $\alpha$-pinene and m-xylene, and ambient secondary aerosol (SA) formation following OH exposure in the PFA OFR. Given the emergence of OFRs as a technique to characterize SA formation, I might support eventual publication of this manuscript in AMT after my comments below are taken into consideration.

**General Comments**

1. There is a growing body of literature suggesting that adding 185 nm irradiation in OFRs – along with 254 nm – is advantageous to use of only 254 nm radiation with externally added ozone, especially with respect to organic peroxy radical ($RO_2$) chemistry, resilience to OH suppression and UV photolysis, and easier operation in the field (e.g. Peng et al., 2019; Peng and Jimenez, 2020; Rowe et al., 2020). It isn't clear to me from the text (L137-L141, L151-L155) if the PFA OFR can implement the OFR185 mode or not. If OFR185 operation is possible, it's worth clarifying that, and explaining why it wasn't evaluated here. If it is not possible, please clarify, and discuss the associated tradeoffs.

2. Overall, the most novel aspect of the PFA OFR design appears to be the higher reflectivity achieved with the ePTFE gasket combined with the lower lamp power. This design modification enables the PFA OFR to achieve a higher OH exposure at a specific lamp power relative to other designs, as noted in L128-L130, which is noteworthy. The potential implications that are identified from the results seem to be better residence time distributions because of less recirculation and reduced temperature gradients. Aside from that, the implication on measurements of interest was less clear. The gas and penetration efficiencies are comparable to previous OFR designs with broader RTDs and less internal reflectivity, as are the $\alpha$-pinene and m-xylene SOA yields. To me, this suggests that results of the sort described here are not sensitive to this design component, or that OFR applications that might be affected by higher internal reflectivity are not adequately discussed. I would strongly encourage adding a section that illustrates applications where this higher reflectivity demonstrably improve performance using metrics other than the OH exposure.

3. The comparison of particle penetration efficiencies is incomplete, and in some places is misleading. Figure 7 shows that the size-dependent particle penetration efficiencies of PFA, PAM and TPOT OFRs, and as presented, suggests that the PFA performance is the best of the three. However, as noted in L215-L220, the PFA was conditioned for 12 hours prior to testing to suppress static discharge, whereas the other OFRs were not. Thus, results are a combination of OFR design and testing procedure, and how to isolate the relative importance of each factor is not clear. Either results for the PFA prior to conditioning also need to be shown for a direct comparison, or this difference needs to be more clearly identified in the figure/caption. Figure 7 also does not show published particle transmission efficiency data for several other OFR designs that were already referenced in this paper. Please see Figure 2 from Li et al., 2019, reproduced below for reference; to my knowledge, this is the most comprehensive comparison to date:

[Figure]

**Figure 2.** Particle (left and bottom axis) and gas (right and top axis) transmission efficiencies ($P_{trans}$ and $G_{trans}$) for the ECCC-OFR. Particle transmission efficiencies of other OFRs are shown for comparison: PAM glass and TPOT (Lambe et al., 2011), PAM metal (Karjalainen et al., 2016), TSAR (Simonen et al., 2017), CPOT (Huang et al., 2017), and PEAR (Ihalainen et al., 2019).

4. Similarly, the gas penetration efficiency may have been measured in different ways. For example, in measurements by Lambe et al. (2011) and Li et al. (2019), the OFR walls were first passivated by flowing the relevant gas(es) through the OFR. Based on the text (L199-L200), it doesn't appear that that was done here, in which case this may be a plausible explanation for the lower $SO_2$ penetration efficiency in the PFA OFR.

5. Occasionally the reactor is referred to as "PFA". It might be less ambiguous to refer to it as the "PFA OFR" to distinguish it from perfluoroalkoxy alkanes.

**Technical Comments**

6. L148-L150: The authors state that "Continuous operation for 6 hours resulted in a temperature rise of less than 2°C" What is the temperature rise over 24 hours or longer, i.e. periods that would be relevant for continuous ambient OFR measurements?.

7. L168: Please mention the OD of the copper bypass line, clarify the reason for using a 150 cm length of bypass inlet versus 200 ccm length of OFR inlet, and calculate the residence time in the bypass and OFR inlet lines to place in context of the OFR residence time.

8. L313: Were different side flow:center flow ratios studied to evaluate the influence of this flow ratio on the residence time distribution? Is a side:center flow ratio of 1:1 optimal, or could the RTD be further improved at a different value?

9. L356: This statement is not correct – the TPOT and PAM OFRs were also operated in OFR254 mode in the study described here.

10. L412-L414, L423-425: Please apply the OFR254 OH exposure estimation equation developed in Section 3.7 of Peng et al. (2015) to calculate the OH exposure during these SOA yield measurements. As far as I can tell, the required inputs to this equation are available from the measurements that were described here.

11. Figure 1 and Section 2.1: Please explain/justify the use of 35° and 24° inlet and outlet cone angles.

12. Figure 3 should either be moved to Supplement or deleted and described briefly in words.

13. Figure 4 could be moved to Supplement.

14. Figure 6a: it would be better to present results in terms of the photon flux, which is an intrinsic property of the OFR that could be more easily compared with other OFR designs, rather than the fractional lamp power, which is only applicable to the specific lamp type used here. The photon flux could be estimated from the maximum lamp output normalized by the internal surface area of the OFR, or, preferably, constrained using ozone measurements measured at the exit of the OFR (as a function of humidity and lamp power) using a photochemical model such as the OFR-KinSim mechanism (Peng and Jinenez, 2019, 2020). Also, please change the y-axis to a logarithmic scale, or make the y-axis scale go to zero.

15. Figure 6b: I don't understand the utility of showing the single point obtained without ePTFE at 50% lamp power. With only 2-3 data points shown here, it might just be easier to integrate this data into Figure 6a.

16. Figure 7 is incomplete (see comment #3).

17. Figure 8: Please clarify whether the literature data shown here were obtained with UV lamps on or off.

18. Figure 9 could be moved to Supplement.

19. Figure 10 could be moved to Supplement.

20. In my opinion, Figure 11 should plot the SOA yield as a function of OH exposure rather than $C_{OA}$. The precursor concentration was not systematically varied, and $C_{OA}$ is not really the independent variable here. The OH exposure can be estimated using the OFR254 estimation equation provided by Peng et al. (2015).

21. Figure 13 could be moved to Supplement.

**References**

Li, K., Liggio, J., Lee, P., Han, C., Liu, Q., and Li, S.-M.: Secondary organic aerosol formation from $\alpha$-pinene, alkanes, and oil-sands-related precursors in a new oxidation flow reactor, Atmos. Chem. Phys., 19, 9715–9731, https://doi.org/10.5194/acp-19-9715-2019, 2019.

Z. Peng, D.A. Day, H. Stark, R. Li, B.B. Palm, W.H. Brune, and J.L. Jimenez. HOx radical chemistry in oxidation flow reactors with low-pressure mercury lamps systematically examined by modeling. *Atmos. Meas. Tech.*, 8, 4863-4890, doi:10.5194/amt-8-4863-2015, 2015.

Z. Peng and J. L. Jimenez. KinSim: A Research-Grade, User-Friendly, Visual Kinetics Simulator for Chemical-Kinetics and Environmental-Chemistry Teaching, *J. Chem. Educ.* 2019, 96, 4, 806–811. https://pubs.acs.org/doi/10.1021/acs.jchemed.9b00033.

Z. Peng, J. Lee-Taylor, J.J. Orlando, G.S. Tyndall, J.L. Jimenez. Organic peroxy radical chemistry in oxidation flow reactors and environmental chambers and their atmospheric relevance. *Atmos. Chem. Phys.,*19, 813-834, 2019, https://doi.org/10.5194/acp-19-813-2019.

Z. Peng, J.L. Jimenez. Radical chemistry in oxidation flow reactors for atmospheric chemistry research. *Chem. Soc. Rev.*, 49, 2570-2616, doi: 10.1039/C9CS00766K, 2020.

Rowe, J. P., Lambe, A. T., and Brune, W. H.: Technical Note: Effect of varying the $\lambda = 185$ and 254 nm photon flux ratio on radical generation in oxidation flow reactors, Atmos. Chem. Phys., 20, 13417–13424, https://doi.org/10.5194/acp-20-13417-2020, 2020.

---

## Editor Comment (EC1) · Mingjin Tang (Editor) · 7 Jan 2021

Dear colleagues,

Should the full name of OFR be included in the title of this manuscript?

Mingjin Tang Handling editor

---

## Author Comment (AC1) · 24 Jan 2021

**Response to Anonymous Referee #1**

The authors present here a description and investigation of a new oxidation flow reactor that is made entirely from Teflon. A variety of OFRs have been developed over the past decade from a range of materials, but few have been previously designed from Teflon despite its high inertness and low vapor wall losses. The specific geometry and design of each has important implications for interpreting the data they produce. It is consequently valuable and important to conduct (and submit for peer-review) the detailed sorts of testing described here, and this is a fitting journal for this work. This manuscript presents a detailed description of the design and examination of its residence times, wall losses, and results, and it should be published. My concern, however, is not so much with the science (though there are some specific comments below), but in the presentation. Overall, I think many of the Figures could be made more clear, and many of them (and some of their discussion) does not need to be included in the main body and could be moved to an SI.

**General comment:**

(1) In the descriptions of the operation of the PFA, the authors spend a lot of time discussed operating modes that are not necessarily pertinent to the final operational approach. For instance, in some figures they show the impacts of operating with the side purge flow off, or without the reflective ePTFE. These feel to me like advancements the authors figured out along the way that don't describe the actual recommended operation of the system (i.e., the authors do not seem to recommend ever operating with the side purge flow off). Consequently, they seem better suited to a brief mention in the main text and deeper discussion in a Supplementary Information for the more interested reader. The discussion of the static removal is similar, though I think may warrant a discussion in the main text as the static losses might be expected to be the main issue with a non-conductive OFR. Figures (and associated discussions )that I think are more detail than are maybe relevant to the main text are: Figure 3 - This figure is illustrative of the detailed description of the text, I don't think it really adds much Figure 5 - shows the importance of the reflective sheath, but since the reflective sheath is used throughout the manuscript and is "part" of the instrument, this mostly shows work done along the way, and is not of central importance to the rest of the work. One option would be to show the gradient by pairing Fig. 5a and Fig 6a as one figure, as Fig 5b (which is just the ratios shown in 5a) and 6b (which doesn't have much information) don't seem critical. Figure 9 - again, as with Figure 5, this mostly shows work done along the way, and is not really germane to the final instrument. The interested reader might want to dig into this, but since the average reader is probably more interested in the final configuration, this could go to the SI

This is a general issue, not just related to these specific figures. Overall, I think discussions of some of the methods are perhaps a bit too detailed, and then interpretation of the demonstration and scientific data at the end is a bit sparse.

Answer: We thank the reviewer for the constructive suggestions on the manuscript. Here we list the changes to the current version:

(1) **Section 2.1 Reactor design** has been divided into three subsections.
(2) Figures 3 and 9 have been moved to the supplement.
(3) We have substantially revised the text describing the importance of the reflective ePTFE layer in **Section 3.1** and also state the importance of the side flow in **Section 3.3**

We prefer to keep Figure 5 in the main text of the modified manuscript because it provides justification for the use of ePTFE gasket and shows the significant improvement in the UV intensity and uniformity achieved when it was added.

**Specific comments:**

**(2) Line 89-90: I am not sure, but I thought there was also a teflon film OFR in use by the Jimenez group at some point? Bill Brune also published one in 2007: https://acp.copernicus.org/articles/7/5727/2007/acp-7-5727-2007.pdf . Given that one of the big values/advances of the present work is the all-Teflon construction, some discussion of Teflon OFRs should be included.**

Answer: We have added a brief description of that Teflon-film OFR (Kang et al., 2007) to the introduction.

**(3) Line 107: It is confusing to call it the PFA. I get that it is a play on the fact that it is made of PFA, but it muddies the discussion somewhat.**

Answer: We obviously chose the name because it's made of PFA and admit that this name is somewhat confused with the PFA material. We replaced "PFA reactor" with "PFA OFR" to make it easier to keep straight. "PFA" without OFR following is used only when referring to the material itself.

**(4) Line 124: In Figure 1 the annulus and pinholes are not really clear, so it took a little bit of re-reading and examining to understand exactly how it is designed and works, and I'm still not totally sure I have it figured out.**

Answer: Thanks for pointing this out. We have clarified the text as follows "The outer ~50 % of the flow that is most influenced by interactions with the reactor walls flows outside of the exit cone and into an annulus surrounding it. From there, it is pulled through 12 uniformly spaced ~0.15 cm ID pinholes drilled through the PFA pipe about 3 cm from the top. The flow extracted through those pinholes travels into a channel between the flow tube and the end cap and then through a port on the top cap where it is purged by a vacuum pump,"

**(5) Line 280: Not really a 36% increase in transmission efficiency, more accurately a 36 percentage point increase, maybe state as "36% of particles that are no longer lost"**

Answer: Thanks for pointing this out. We have rewritten the sentence "The similarity in the resulting 36% of the 50 nm particles that are no longer lost and the 41% of those 50 nm particles that are expected to be charged (Wiedensohler, 1988) suggests electrostatic loss was negligible after the static charge was minimized."

**(6) Line 284: Not clear, is the data in Figure 7 directly from the referenced work?**

Answer: Yes, the data are directly from the referenced work and we revised the figure and added the most recent published particle transmission efficiency data for several other OFR designs. Please refer to Figure 6 in the revised paper and to the response to Anonymous Reviewer #2 general comment (2) for details.

**(7) Line 309: I'm not a statistician, so I'm not sure whether or not this is the right metric or approach – why adjust residence times by the average? Why not normalized to the nominal residence time (V/Q)? Isn't that the metric that is used for the laminar flow case? For example, in Figure 8b, SO2 comes out sooner than expected from laminar flow or than CO2, which the authors attribute to reversible uptake – but that should delay the SO2 response, not speed it up. By delaying response, the average RT is shifted later, and the apparent speed of the early eluting SO2 is faster. I think all of this would be easier to interpret (and perhaps more meaningful) if the nominal RT were used instead of the average.**

Answer: We appreciate the noted advantage of using V/Q rather than an average that is calculated from the measurements.  One complication for this OFR is that the residence time of the subsampled central flow should differ from that expected from V/Q because the velocity in the laminar flow profile is greatest in the center of the tube.  In addition to the offset that would result, different sample to purge flow ratios were used for different tests, which would cause shifts in the distributions unrelated to effects such as wall interactions.  Perhaps more importantly, we normalize by the average calculated from the data because that the approach that is used to present similar experimental data for the other OFRs (Huang et al., 2017, Lambe et al., 2011), which facilitates comparisons.

**(8) Lines 342-361: How should the scientific community be dealing with or thinking about these yields? Looking at Figure 11, OFRs seem to be saying that the yield for a-pinene at a give Coa and OHexp could easily vary by a factor of many (e.g. ~5-50% for Coa = ~100ug/m3 and OH in the range of 3-10x10^11). This is, of course, not a problem the authors are solely responsible for, but it makes me wonder about the utility of using yield comparisons to validate or understand the OFR. It is reassuring, I suppose, that the PFA is in the middle of all this noise (as opposed to outlying), but are we (the reader and/or the authors) really learning anything from this intercomparison? Is there some way to make sense of this spread other than the broad "uncertainty in OHexp"? Maybe the points in Figure 11 could at least indicate different OFR modes, or something that might explain some of these differences?**

Answer: We agree.  There is, of course, variability in yields reported from smog chambers as well, though not as much as is evident for OFRs in the figure.  We attribute some of the variation to differences in gas and particle wall losses and to the uncertainty in OHexp noted by the reviewer.  Because the comparison is with OFRs that have undergone much more extensive use and characterization than that described here, our primary objective was to show that results with this OFR are comparable to those from others.  We hope to investigate run-to-run and OFR-to-OFR variability more in the future.

**(9) Line 366: In Figure 12 it looks like the seed is multi-modal, but the test refers to a narrow mode at 200 nm - is this due to multiple charging in the DMA?**

Answer: Yes, this is due to multiple charging. The multiple charge modes are pronounced because a volume (rather than number) concentration size distribution is presented. The 200 nm mode would be more amplified relative to the multiple charge modes if the surface area concentration or condensation sink distribution was instead presented.

**(10) Line 374-376: Adding a panel to Figure 12 showing the trend in yield as a function of pinene:AS would aid this discussion**

Answer: Thanks for the recommendation.  We added **Figure S3** that shows the trend in yield as a function of precursor:seed ratio.

**(11) Line 384: Both of these citations refer to specific conditions and locations in the early 2000's. How are the author's using these to estimate the stated OHR?**

Answer: We only used this as a rough guide for OHR based on prior measurements in CA because we didn't measure it during the ambient measurement period. We agree that the OHR is considerably lower today than in the early 2000's.  We have removed the statement in the manuscript.

**Figure comments:**

**(12) Figure 2 is missing "(a)"**

Answer: Corrected.

**(13) Figure 3 labels are confusing because the arrows could refer to "that point onward" or to the region they are coming from. It would be clearer to indicate with colored regions or a binary on/off trace when UV was on or off.**

Answer: Thanks for pointing this out. We moved it to supplement and added a binary on/off trace when UV was on/off. Please refer to **Figure S1** in the revised paper.

**(14) Figure 6 could be made more clear - too much going on in the legend with lots of repetition. Also not clear if 6a is with or without the ePTFE. I gather it is without? Why are only two points from 6a reproduced on 6b? Why not just include the one point w/ ePTFE shown in 6b into 6a and make all the labeling more clear?**

**Answer:** Thanks for pointing this out. We have combined **Figures 6 (a) and (b)** and show the enhancement in intensity/photon flux achieved by adding the ePTFE gasket. Please see **Figure 5** in the revised paper.

**(15) Figure 9 - why not also include the modeled with the side flow on?**

Answer: Our goal was only to show that the experimental results were consistent with the simulations. We did not include a similar comparison for operation with the side flow on simply because our CFD software did not have the package required to do so. However, we added a series of RTDs with different side:main flow ratios to the figure and moved it to the supplement. It is now **Figure S2**.

**(16) Figure 10 legends are similarly busy and repetitive requires a lot of looking back and forth. Some estimate of OH exposure on these plots might help.**

Answer: Thanks for pointing this out. We calculated OH exposure from the equation developed by Peng et al. (2015) and added it in the revised **Figure 8**.

**(17) Figure 11 - make the legend markers darker so they can be seen more easily**

Answer: We have adjusted the contrast of the legend markers. Please refer to **Figure 9** in the revised paper.

---

## Author Comment (AC2) · 24 Jan 2021

Reply to comments by Anonymous Referee #2

**General Comments**

1. **There is a growing body of literature suggesting that adding 185 nm irradiation in OFRs – along with 254 nm – is advantageous to use of only 254 nm radiation with externally added ozone, especially with respect to organic peroxy radical (RO2) chemistry, resilience to OH suppression and UV photolysis, and easier operation in the field (e.g. Peng et al., 2019; Peng and Jimenez, 2020; Rowe et al., 2020). It isn't clear to me from the text (L137-L141, L151-L155) if the PFA OFR can implement the OFR185 mode or not. If OFR185 operation is possible, it's worth clarifying that, and explaining why it wasn't evaluated here. If it is not possible, please clarify, and discuss the associated tradeoffs.**

   Answer: The PFA OFR is operated only in 254 mode. An explanation was provided in what is now Section 2.1.2:

   "Though the combination of materials results in sufficiently high reflectance for the 254 nm emission peak of a mercury lamp. Silva et al. (2010) showed that the reflectance of ePTFE at 175 nm is significantly lower, with the difference thought to be due to absorption by O2 trapped in pores. Reflectance at the 185 nm emission peak of a mercury lamp is expected to be slightly higher than that at 175 nm, but it is likely that a significant intensity gradient would still exist and so a 254 nm-only lamp is used and ozone generated externally and introduced with the sample flow."

   Nevertheless, we appreciate the advantages of operation with 185 and 254 nm and will continue to evaluate design modifications that might enable it with our OFR.

2. **Overall, the most novel aspect of the PFA OFR design appears to be the higher reflectivity achieved with the ePTFE gasket combined with the lower lamp power. This design modification enables the PFA OFR to achieve a higher OH exposure at a specific lamp power relative to other designs, as noted in L128-L130, which is noteworthy. The potential implications that are identified from the results seem to be better residence time distributions because of less recirculation and reduced temperature gradients. Aside from that, the implication on measurements of interest was less clear. The gas and penetration efficiencies are comparable to previous OFR designs with broader RTDs and less internal reflectivity, as are the α-pinene and m-xylene SOA yields. To me, this suggests that results of the sort described here are not sensitive to this design component, or that OFR applications that might be affected by higher internal reflectivity are not adequately discussed. I would strongly encourage adding a section that illustrates applications where this higher reflectivity demonstrably improve performance using metrics other than the OH exposure.**

   Answer: We thank the referee for the recommendation. We have substantially revised the text describing potential application in **Sections 2.1.2 and 2.1.3**, the importance of the

reflective ePTFE layers in **Section 3.1,** and the importance of the side flow **in Section 3.3.** Please refer to the revised paper.

**The comparison of particle penetration efficiencies is incomplete, and in some places is misleading. Figure 7 shows that the size-dependent particle penetration efficiencies of PFA, PAM and TPOT OFRs, and as presented, suggests that the PFA performance is the best of the three. However, as noted in L215-L220, the PFA was conditioned for 12 hours prior to testing to suppress static discharge, whereas the other OFRs were not. Thus, results are a combination of OFR design and testing procedure, and how to isolate the relative importance of each factor is not clear. Either results for the PFA prior to conditioning also need to be shown for a direct comparison, or this difference needs to be more clearly identified in the figure/caption. Figure 7 also does not show published particle transmission efficiency data for several other OFR designs that were already referenced in this paper. Please see Figure 2 from Li et al., 2019, reproduced below for reference; to my knowledge, this is the most comprehensive comparison to date:**

[Figure]

**Figure 2.** Particle (left and bottom axis) and gas (right and top axis) transmission efficiencies ($P_{trans}$ and $G_{trans}$) for the ECCC-OFR. Particle transmission efficiencies of other OFRs are shown for comparison: PAM glass and TPOT (Lambe et al., 2011), PAM metal (Karjalainen et al., 2016), TSAR (Simonen et al., 2017), CPOT (Huang et al., 2017), and PEAR (Ihalainen et al., 2019).

Answer: We added the data for the CPOT, TSAR, and ECCC-OFR to the revised **Figure 6**. The results are not meant to be misleading. Our intent is to always minimize charge prior to

measurements. Though we have not tested this extensively yet, it is our belief that the charge will not return until and unless the OFR is disassembled or relocated.

3. **Similarly, the gas penetration efficiency may have been measured in different ways. For example, in measurements by Lambe et al. (2011) and Li et al. (2019), the OFR walls were first passivated by flowing the relevant gas(es) through the OFR. Based on the text (L199-L200), it doesn't appear that that was done here, in which case this may be a plausible explanation for the lower SO$_2$ penetration efficiency in the PFA OFR.**

Answer: Thanks for pointing this out. We did first passivate the reactor by flowing the relevant gas through it for at least 15 minutes before the experiment and until the concentration measured at the outlet was constant. Even so, the SO$_2$ penetration efficiency was still relatively low compared with the others shown. We are unsure why the penetration efficiency is lower and plan to investigate this further.

**Occasionally the reactor is referred to as "PFA". It might be less ambiguous to refer to it as the "PFA OFR" to distinguish it from perfluoroalkoxy alkanes.**

Answer: Thank you for the recommendation, which we have followed. Please refer to the response to Anonymous Reviewer #1 general comment (3) for details.

**Technical Comments**

6. **L148-L150: The authors state that "Continuous operation for 6 hours resulted in a temperature rise of less than 2 $^{\mathrm{O}}$C" What is the temperature rise over 24 hours or longer, i.e. periods that would be relevant for continuous ambient OFR measurements?.**

Answer: We didn't observe significant temperature increase during the 3-day ambient measurements. We added this statement in the revised paper.

7. **L168: Please mention the OD of the copper bypass line, clarify the reason for using a 150 cm length of bypass inlet versus 200 ccm length of OFR inlet, and calculate the residence time in the bypass and OFR inlet lines to place in context of the OFR residence time.**

Answer: We mentioned the diameter (0.635 cm OD) in section 2.4. The residence time of the bypass is approximately 2 s. We added a sentence about it in the revised paper.

8. **L313: Were different side flow:center flow ratios studied to evaluate the influence of this flow ratio on the residence time distribution? Is a side:center flow ratio of 1:1 optimal, or could the RTD be further improved at a different value?**

Answer: We did some measurements to evaluate the influence of the flow ratio on the residence time, with the results added to **Figure S2** in the revised paper. It may be that increasing the ratio beyond 1:1, especially if accompanied by a reduction in the cross section of the exit cone, would result in narrowing of the RTD. However, that benefit would have to be balanced with the resulting decrease in sample flow rate or average residence time.

9. **L356: This statement is not correct – the TPOT and PAM OFRs were also operated in OFR254 mode in the study described here.**

Answer: Thanks for pointing this out. We have removed this statement in the revised paper.

10. **L412-L414, L423-425: Please apply the OFR254 OH exposure estimation equation developed in Section 3.7 of Peng et al. (2015) to calculate the OH exposure during these SOA yield measurements. As far as I can tell, the required inputs to this equation are available from the measurements that were described here.**

Answer: We added the OH exposure estimation in the SOA yield measurements. Please refer to **Section 3.4** in the revised paper.

11. **Figure 1 and Section 2.1: Please explain/justify the use of $35^0$ and $24^0$ inlet and outlet cone angles.**

Answer: We wanted to keep the angles close to the 30 degrees suggested by Huang et al. (2017), but they were also constrained by the diameters of the PFA tube and the exit flow port and by the capabilities of the machine shop we used.. We added a statement about this in **Section 2.1.1**.

12. **Figure 3 should either be moved to Supplement or deleted and described briefly in words.**

Answer: Done.

13. **Figure 4 could be moved to Supplement.**

Answer: We moved it to **Section 2.1.1**.

14. **Figure 6a: it would be better to present results in terms of the photon flux, which is an intrinsic property of the OFR that could be more easily compared with other OFR designs, rather than the fractional lamp power, which is only applicable to the specific lamp type used here. The photon flux could be estimated from the maximum lamp output normalized by the internal surface area of the OFR, or, preferably, constrained using ozone measurements measured at the exit of the OFR (as a function of humidity and lamp power) using a photochemical model such as the OFR- KinSim mechanism (Peng and Jinenez, 2019, 2020). Also, please change the y-axis to a logarithmic scale, or make the y-axis scale go to zero.**

    Answer: Thanks for the suggestion. We moved Figure 6 (a) to the supplement and combined Figure 6 (b) and Figure 6 (a). The recommended change was made in what is **Figure 5** in the revised paper.

15. **Figure 6b: I don't understand the utility of showing the single point obtained without ePTFE at 50% lamp power. With only 2-3 data points shown here, it might just be easier to integrate this data into Figure 6a.**

    Answer: Please see the response to the previous comment.

16. **Figure 7 is incomplete (see comment #3).**

    Answer: Done.

17. **Figure 8: Please clarify whether the literature data shown here were obtained with UV lamps on or off.**

    Answer: Done, please refer to **Figure 7** in the revised paper.

18. **Figure 9 could be moved to Supplement.**

Answer: Done.

19. **Figure 10 could be moved to Supplement.**

Answer: We revised this figure and pointed out the estimated OH during the SOA yield measurement.  Please refer to **Figure 8** in the revised paper.

**20. In my opinion, Figure 11 should plot the SOA yield as a function of OH exposure rather than C$_{OA}$. The precursor concentration was not systematically varied, and C$_{OA}$ is not really the independent variable here. The OH exposure can be estimated using the OFR254 estimation equation provided by Peng et al. (2015).**

Answer: We added a sentence in the figure caption to point out that the color represents the OH exposure, which was estimated from the OFR254 estimation equation provided by Peng et al. (2015). We didn't test the SOA yield over a wide range in OHexp, which is why we prefer to use C$_{OA}$ as the x-axis. It also simplifies comparison with the other OFR studies presented in Lambe et al. (2011). Please see **Figure 9** in the revised paper.

**21. Figure 13 could be moved to Supplement.**

Answer: Though we moved some figures to the supplement, we prefer to keep this figure in the main text because it shows that the rapid particle concentration changes in the sampling period can be captured by the PFA OFR, which is one focus of our future studies.

---

## Author Comment (AC3) · 24 Jan 2021

1. **I think in the Introduction part it is worth mentioning that the PFA is vertically oriented and the heating source is on the top, thus minimizing the convection and maintaining laminar flow profile, which is different from other oxidation flow reactors.**

Answer: Thanks for pointing this out. We added those details in the introduction.

2. **Section 2.1 is a little difficult to follow and it does take me a while to figure out the actual design. I think it needs to be rearranged with subsections, e.g., inlet, main tube, outlet, lamps, etc., to make it clearer.**

Answer: Please refer to the response to Anonymous Reviewer #1 general comment (1) for details of changes made to the manuscript structure.

3. **Line 121: based on the parabolic flow assumption and the dimension (2.1 cm ID of the cone and 7.8 cm ID of the tube), it is just about 14% flow that goes into the cone ideally. That 50% flow is extracted at the cone suggests that it is not the parabolic profile at the outlet. Maybe clarify this point here.**

Answer: Sorry for the mistake, we corrected this in the revised paper. The ID of the exiting cone should be 4 cm, with the result that the roughly 50:50 flow split does not result in significant deviation in gas streamlines at the exit.

4. **About the lamp: if I understand correctly, the lamp is exposed directly to the flow? Also, since Sections 2.6 & 3.2 presented the experimental measurement of light intensity as a function of the distance to the bottom, it is worth mentioning it in Section 2.1 at least or rearranging the two sections.**

Answer: Yes, the lamp is exposed directly to the flow, but only the side flow after it is split from the sample flow. So there is no contact between the sample flow and the lamp surface. We added a sentence clarifying this in the revised paper.

5. **About the position of ePTFE: is it inside the PFA tube or outside the PFA tube but inside the layer of aluminized Mylar?**

Answer: The latter – outside the PFA tube but inside the layer of aluminized Mylar.

6. **Line 142: it is not very clear how the two U-bolts hold the tube inside the aluminum tube.**

Answer: Those two U-bolts are mounted through opposite sides of the square enclosure as shown in the top view sketch below.

[Figure]

7. **Line 148: is the temperature referring to the gas coming out of the PFA tube or that coming between the shell?**

Answer: We measured the temperature on the outside surface of the PFA tube during the laboratory experiments. The warmest point, which is close to the lamp, was stable at 23.6 C. We believe the sample flow temperature lies between that and room temperature because it only experiences a small amount time around the warm outlet.

8. **About the experimental setup: how is the gas getting through the tube, by pulling due to vacuum at the outlet or pushing at the inlet? For each case, there should be an excess flow (or open port), but Fig. 2 cannot tell. I guess it is pulled by the vacuum at the outlet?**

Answer: Yes, it's pulled by the vacuum pump at the outlet; we didn't control the inlet flow.

9. **About purge flow: does this flow determine the actual flow fraction that goes into the outlet cone?**

Answer: Yes, it does. We control the total flow to be 3 lpm. The sample through the center port was connected to an SMPS that had a fixed flow rate of 1.5 lpm. The side flow was extracted by a vacuum pump and was controlled to 1.5 lpm using a rotameter.

**10. About the charge effect on particle wall loss: does it have a preference on a specific polarity when there is a static charge on the surface?**

Answer: We did not evaluate this but expect that it would be comparable for positively and negatively charged particles for reasons similar to those described by McMurry and Rader (1985) for Teflon smog chambers.

**11. Line 221: how is the gas transmission efficiency determined by pulse injection? Does the concentration downstream refer to integrated concentration?**

Answer: The gas transmission efficiency was determined through steady state experiments in which the gas was continuously injected. We revised this sentence to make it clearer. Please see **Section 2.4** in the revised paper.

**12. Section 3.1: same as comment 3, how does the simulated velocity distribution look like at the outlet w/ and w/o purge flow?**

As noted in the response to comment 3, the exit port diameter stated in the original manuscript was incorrect. The velocity distribution is expected to be minimally perturbed in the exit region with the purge flow on.

**13. Figures 5 & 6 show that even with the high-reflective layer, the UV intensity at the inlet can be one-third of that at the outlet and the corresponding OH concentrations could be the same ratio. Thus there could exist a transition regime in the reactor that a compound reacts with $O_3$ and OH equally. Though it does not affect the OH exposure calculation, knowing the distributions of $O_3$ and OH along the tube will be helpful to analyze the competing reaction channels. If this point is out of the scope of this manuscript, it is worth mentioning that point though.**

Answer: Done. Please see **Section 2.5** in the revised paper.

**14. For the transmission efficiency of sticky vapor molecules, I am wondering if the transmission efficiency increases as the Teflon wall gets saturated after absorption/adsorption for a long time? The same question goes to Section 3.5, will the SOA yield increase after the experiment runs for a long time when the Teflon wall gets saturated with the sticky vapor molecules? For experiments with seeds, what is the timescale loss to the wall versus that to the particle surfaces?**

Answer: We didn't observe transmission efficiency increases with time. We conducted the experiments for at least 8 hours and each was repeated at least 3 times; no increase in transmission efficiency was evident. The same is true for the SOA yield results reported in Section 3.5. Again, we didn't observe any change over time or between repeat experiments. We have not yet estimated the relative timescales of vapor loss to the walls and to particle surfaces, in part because much of our initial effort involved variation of parameters that would produce a wide range in the resulting balance between the two.

**15. Equation 3: since RTD is the term used in the context, PDF may be replaced with RTD.**

Answer: We would like to keep this to be consistent with use in other OFR papers and the RTD comparison provided in the PAM wiki website.

**16. Line 324: what is the purpose of turning off the purge flow?**

Answer:  It will always be on during normal operation.  It was turned off only to provide a contrast that demonstrates its impact.

**17. Lines 351 and 360: it is worth noting here that the reactivities of OH and VOC are different.**

Answer: Thanks for the comment.

**18. Line 384: How is the ambient OHR determined?**

Answer: In response to Anonymous Reviewer #1 general comment (11), we removed the reference reporting OHR in the study region because it is likely not representative of present day conditions.  No gas phase measurements were made during the tests and so we have no way of constraining the OHR.

**19.  Figure 10: For experiments under the same condition, how are the different data points from since the flow reactor is usually running at a steady-state? If it is because of different injected VOC concentrations, it is better to mention that in the caption.**

Answer: Yes, the different data points were because the injected VOC concentrations were different.  We clarified this in the caption of what is now **Figure 8** in the revised paper

---

## Author Comment (AC4) · 24 Jan 2021

1. Should the full name of OFR be included in the title of this manuscript?

Answer: Thanks for the recommendation. We retitled the paper "Design and Characterization of a new OFR for laboratory and long-term ambient studies"

---

## Referee Report (RR1)

RC3 #2: Overall, the most novel aspect of the PFA OFR design appears to be the higher reflectivity achieved with the ePTFE gasket combined with the lower lamp power. This design modification enables the PFA OFR to achieve a higher OH exposure at a specific lamp power relative to other designs, as noted in L128-L130, which is noteworthy. The potential implications that are identified from the results seem to be better residence time distributions because of less recirculation and reduced temperature gradients. Aside from that, the implication on measurements of interest was less clear. The gas and penetration efficiencies are comparable to previous OFR designs with broader RTDs and less internal reflectivity, as are the α-pinene and m-xylene SOA yields. To me, this suggests that results of the sort described here are not sensitive to this design component, or that OFR applications that might be affected by higher internal reflectivity are not adequately discussed. I would strongly encourage adding a section that illustrates applications where this higher reflectivity demonstrably improve performance using metrics other than the OH exposure.

Author response: We thank the referee for the recommendation. We have substantially revised the text describing potential application in Sections 2.1.2 and 2.1.3, the importance of the reflective ePTFE layers in Section 3.1, and the importance of the side flow in Section 3.3. Please refer to the revised paper.

Reviewer response: The text that the authors added to Sections 2.1.2, 2.1.3, 3.1, and 3.3. provides useful additional details about the design advantages. It is clear that the RTD is improved in the PFA OFR. However, it is still not clear to me which OFR applications are significantly affected by these design advantages - even with their implementation, the effect on gas/particle penetration efficiency and SOA yields is minor at best. My interpretation of this result is that gas/particle penetration efficiencies and SOA yields are not very sensitive to the RTD. I think the paper would be more compelling if they can present results and/or describe OFR applications that are more clearly affected by the improved RTD than SOA yields and gas/particle transmission.

---

## Author Response (AR2)

**My comments on the revised manuscript are in red text below.**

Please swap the axes in Figure 5 of the revised paper so that the photon flux is on the x-axis and the OH exposure is on the y-axis.

Answer: We thank the referee for the recommendation. We reversed the axes in Fig. 5; please refer to the revised paper.

RC3 #2: Overall, the most novel aspect of the PFA OFR design appears to be the higher reflectivity achieved with the ePTFE gasket combined with the lower lamp power. This design modification enables the PFA OFR to achieve a higher OH exposure at a specific lamp power relative to other designs, as noted in L128-L130, which is noteworthy. The potential implications that are identified from the results seem to be better residence time distributions because of less recirculation and reduced temperature gradients. Aside from that, the implication on measurements of interest was less clear. The gas and penetration efficiencies are comparable to previous OFR designs with broader RTDs and less internal reflectivity, as are the α-pinene and m-xylene SOA yields. To me, this suggests that results of the sort described here are not sensitive to this design component, or that OFR applications that might be affected by higher internal reflectivity are not adequately discussed. I would strongly encourage adding a section that illustrates applications where this higher reflectivity demonstrably improve performance using metrics other than the OH exposure.

Author response: We thank the referee for the recommendation. We have substantially revised the text describing potential application in Sections 2.1.2 and 2.1.3, the importance of the reflective ePTFE layers in Section 3.1, and the importance of the side flow in Section 3.3. Please refer to the revised paper.

Reviewer response: The text that the authors added to Sections 2.1.2, 2.1.3, 3.1, and 3.3. provides useful additional details about the design advantages. It is clear that the RTD is improved in the PFA OFR. However, it is still not clear to me which OFR applications are significantly affected by these design advantages - even with their implementation, the effect on gas/particle penetration efficiency and SOA yields is minor at best. My interpretation of this result is that gas/particle penetration efficiencies and SOA yields are not

very sensitive to the RTD. I think the paper would be more compelling if they can present results and/or describe OFR applications that are more clearly affected by the improved RTD than SOA yields and gas/particle transmission.

Answer: We thank the referee for the recommendation. We identified at least one application for which the narrow RTD is beneficial in the summary, by adding ", making it better suited for measurements of dynamic sources with time-varying composition or concentration." to the end of the sentence that was "Computational simulation and experimental verification of particle and gas residence time distributions (RTDs) show that the flow through the reactor is nearly laminar, with narrower RTDs than reported for OFRs with greater diameter-to-length ratios."

More significantly, we conducted additional yield experiments to quantify the impact of subsampling of the central core flow. A description of the experiments and results is provided in Section 3.4 and the results are summarized in a new Figure S3. As noted in the added text, sampling all of the flow and not just that in the central core results in a broadened RTD and sampling of air that, on average, interacted more with the flow tube walls. The resulting yield curves show that the narrower RTD and reduced wall effects accompanying subsampling of the central flow result in higher SOA yields, which may partially explain why the yield presented in Fig. 9 is slightly higher than that reported for other OFRs.